# The *Aspergillus fumigatus* UPR is variably activated across nutrient and host environments and is critical for the establishment of corneal infection

Manali M. Kamath[1], Jorge D. Lightfoot[1], Emily M. Adams[2], Ryan M. Kiser[3], Becca L. Wells[1], Kevin K. Fuller [1,2]*

1 Department of Microbiology & Immunology, University of Oklahoma Health Sciences Center, Oklahoma City, Oklahoma, United States of America, 2 Department of Ophthalmology, University of Oklahoma Health Sciences Center, Oklahoma City, Oklahoma, United States of America, 3 Department of Pediatrics, University of Oklahoma Health Sciences Center, Oklahoma City, Oklahoma, United States of America

* Kevin-Fuller@ouhsc.edu

**Data Availability Statement:** All relevant data are available within the manuscript and its supporting information files.

## Abstract

The *Aspergillus fumigatus* unfolded protein response (UPR) is a two-component relay consisting of the ER-bound IreA protein, which splices and activates the mRNA of the transcription factor HacA. Spliced *hacA* accumulates under conditions of acute ER stress *in vitro*, and UPR null mutants are hypovirulent in a murine model of invasive pulmonary infection. In this report, we demonstrate that a *hacA* deletion mutant (*ΔhacA*) is furthermore avirulent in a model of fungal keratitis, a corneal infection, and an important cause of ocular morbidity and unilateral blindness worldwide. Interestingly, we demonstrate that *A. fumigatus hacA* is spliced in infected lung samples, but not in the cornea, suggesting the amount of ER stress experienced by the fungus varies upon the host niche. To better understand how the UPR contributes to fungal cell biology across a spectrum of ER-stress levels, we employed transcriptomics on the wild-type and *ΔhacA* strains in glucose minimal media (low stress), glucose minimal media with dithiothreitol (high stress), and gelatin minimal media as a proxy for the nutrient stress encountered in the cornea (mid-level stress). These data altogether reveal a unique HacA-dependent transcriptome under each condition, suggesting that HacA activity is finely-tuned and required for proper fungal adaptation in each environment. Taken together, our results indicate that the fungal UPR could serve as an important antifungal target in the setting of both invasive pulmonary and corneal infections.

## Author summary

Fungal keratitis has emerged as a leading cause of ocular morbidity and unilateral blindness worldwide. Relative to other infectious contexts, however, little is known about the fungal genes or pathways that regulate invasive growth and virulence in the corneal environment. In this report, we demonstrate that genetic disruption of the *Aspergillus fumigatus* unfolded protein response (UPR) abolishes the ability of the mold to establish

**Funding:** This work was supported by the National Institutes of Health (NIH) and Research to Prevent Blindness (RPB). R01EY021725 (NIH) provided salary support to KKF, MMK and EMA; P20GM134973 (NIH) provided salary support to KKF, MMK, BLW, and EMA; an RPB Career Development Award provided salary support to JDL, BLW and KKF; and T32EY023202 (NIH) provided fellowship support to JDL. Research support to the OUHSC Department of Ophthalmology was further provided through P30EY021725 (NIH) and an unrestricted grant from RPB. The funders had no role in the study design, data collection and analysis, decision to publish, or preparation of the manuscript.

**Competing interests:** The authors have declared that no competing interests exist.

infection in a mouse model of FK. Despite this critical role for virulence, however, we did not detect a concerted activation of the pathway beyond levels observed on standard medium, suggesting that the host environment is not an acute source of endoplasmic reticulum stress. Transcriptomic profiling of the wild-type and UPR-deficient strains under host-relevant nutrient conditions revealed a critical role for the pathway in regulating primary and secondary metabolism, cell wall biology, and mitochondrial function, all of which likely modulate fungal growth within and interactions with the host. These results expand our understanding of UPR regulation and function in this important mold pathogen and suggest the pathway could serve as a target for novel antifungals that improve visual outcomes in the setting of fungal keratitis.

## Introduction

The pathogenic potential of *Aspergillus fumigatus* and other fungi is tied to the functional integrity of the endoplasmic reticulum (ER). First, the organelle serves as the synthetic hub for a myriad of cell surface or secreted virulence proteins, including 1) cell wall remodeling enzymes that facilitate apical extension and stress resistance, 2) hydrolases that promote nutrient acquisition from, and penetration through host tissue, and 3) transporters involved in macro/micronutrient uptake or efflux [1–5]. Second, numerous stresses encountered within the host environment – e.g., hypoxia, oxidative stress, or nutrient limitation that places a demand on hydrolase secretion – may cumulatively lead to a cytotoxic aggregation of unfolded peptides in the ER that will induce cell death if left unchecked [2,6,7]. The unfolded protein response (UPR) is an evolutionarily conserved signaling pathway that senses ER stress and drives a transcriptional response that restores protein folding and secretory homeostasis [2,8,9]. Thus, by virtue of its role in ER function, the UPR is a de facto regulator of virulence and a putative target for novel antifungal therapy.

The UPR of *A. fumigatus* is a two-component relay that generally follows the canonical pathway described in *Saccharomyces cerevisiae* [10–12]. The sensing module, IreA, is an ER transmembrane protein whose luminal domain interacts with the protein chaperone BipA/Kar2 under homeostatic conditions. BipA dissociates upon the accumulation of unfolded proteins in the lumen, leading to an oligomerization of IreA molecules that promotes the trans-autophosphorylation of its cytosolic kinase domain and activation of its endoribonuclease domain. The latter splices a 20 bp unconventional intron from the cytosolic *hacA* transcript (*hacA*u, uninduced), leading to a frame-shifted isoform (*hacA*i, induced) that encodes a bZIP transcription factor that provides output for the UPR [13,14]. Work from Askew and colleagues demonstrated that 1) *hacA*i is the most abundant splice form in *A. fumigatus* following treatment with the thiol-reducing agent dithiothreitol (DTT) and, 2) treatment with DTT or tunicamycin, the latter of which inhibits protein glycosylation and normal peptide folding, leads to the induction of ~60 genes in an IreA and HacA dependent manner [15]. This "inducible UPR" (iUPR) consists primarily of genes that promote protein folding (e.g. chaperones, foldases, protein glycosylases) and vesicle trafficking, which is consistent with known UPR-regulated genes in other fungi [15]. The *A. fumigatus* UPR also regulates the expression of calcium ATPases that facilitate the influx of $Ca^{2+}$ ions needed for chaperone function [16]. It follows, therefore, that both *ireA* or *hacA* deletion mutants are hypersensitive to stressors that acutely unfold proteins (high temperature, DTT, TM) or block the egress of proteins from the ER (brefeldin A) [16–18]. The *ΔhacA* or *ΔireA* strains are furthermore growth defective on protein-rich media, such as skim milk agar, which correlates with a lack of detectable

collagenase activity from culture supernatants [14]. This suggests that the enhanced secretory burden experienced on polymeric substrates, and putatively in the lung, is another source of ER stress that promotes increased *hacA* splicing and UPR induction.

Although *hacA^u* is the dominant splice form under homeostatic conditions (e.g. rich media), there is nevertheless a detectable *hacA^i* product at all times. This tracks with a slight growth and conidiation defect of the *ΔhacA* or *ΔireA* strains on standard lab media and suggests that HacA buffers low-level ER stress associated with normal growth and developmental processes. Interestingly, the microarray study by Feng et al. identified ~250 genes that are regulated by IreA and HacA on a rich medium, only 9 of which overlapped with the iUPR defined after DTT and TM treatment [15]. This so-called "basal UPR" (bUPR) includes genes enriched in oxidative metabolism and mitochondrial function, which is distinct from the enriched gene categories associated with the iUPR. One possible explanation is that the bUPR and iUPR are two ends of an activity spectrum that is fine-tuned at the level of *hacA* splicing. In this scenario, UPR gene expression is solely dependent on the canonical HacA protein encoded by the *hacA^i* mRNA. Another possibility is that the bUPR is qualitatively distinct and driven by a protein product encoded by the *hacA^u* mRNA [19]. To date, the UPR-dependent transcriptome of *A. fumigatus* has only been evaluated under conditions of low (rich medium) or acute (e.g. DTT) ER stress, and so a more intermediate level UPR activation and output, if it exists, has yet to be interrogated.

The importance of the UPR in *A. fumigatus* pathogenesis has been evaluated in several murine models of invasive pulmonary aspergillosis (IPA), where *ΔhacA* displays an approximately 50% reduction in virulence based on cumulative mortality [14]. The role of HacA in this regard, as eluded to above, is likely multifactorial. Individual phenotypes associated with *ΔhacA* – including reduced hydrolase secretion, a hypersensitivity to iron limitation [20,21] and cell wall stress [22–25] each have a presumed or demonstrated role in fungal adaptation to the lung environment and probably contribute additively to the virulence phenotype. Interestingly, the *ΔireA* mutant is avirulent in the same infection model. This is consistent with the fact that *ΔireA* displays a greater growth defect at 37°C and an increased sensitivity to iron and nutrient limitation (relative to *ΔhacA*). While this supports a model in which IreA has yet-to-be-identified functions beyond *hacA* splicing, the virulence of *ΔireA* could be restored to wild-type levels by introducing a constitutively spliced *hacA* (*hacAi*) allele [15]. While this does indicate a specific role for the iUPR in supporting *A. fumigatus* virulence, *hacA* splicing within infected tissue has not been formally demonstrated to the best of our knowledge.

Beyond the lung, *A. fumigatus* is also an important causative agent of fungal keratitis (FK), a sight-threatening infection of the cornea that affects 1–2 million individuals globally [26]. The disease occurs when fungal spores or hyphal fragments gain entry to the corneal tissue following damage to the protective epithelium, predominantly as a result of agriculture-related trauma or contact lens wear [27,28]. In all contexts, however, the combination of invasive hyphal growth and leukocytic infiltration causes acute pain, photophobia, and a disruption to the optical properties of the cornea that results in vision loss [29–31]. FK results in the need for corneal transplantation in ~40% of cases and a wholesale removal of the eye in about 10% [26]. As with IPA, the poor outcomes of FK are due, in part, to the inadequacy of current antifungals, for which only the polyene natamycin has received FDA-approved for corneal use [32,33]. Thus, it is a major goal of our group to identify *A. fumigatus* pathways that support virulence in both the lung and corneal environments, allowing for the development of novel antifungals with dual use. The cornea is avascular, immune-privileged, and beyond a thin epithelial and endothelial layer, the bulk of its structure (the central stroma) is a dense extracellular matrix (ECM) of collagen and other proteoglycans. Although these features distinguish the corneal environment from that of the lung, we hypothesized that the fungal UPR would

regulate virulence in both environments due to shared ER stressors, including nutrient limitation or oxidative stress derived from infiltrating leukocytes.

In this study, we demonstrate that an *A. fumigatus* Δ*hacA* mutant is avirulent in a murine model of IPA and FK, suggesting the pathway is activated and functions similarly in the pulmonary and corneal microenvironments. Interestingly, however, we note that the degree of fungal *hacA* splicing differs based on the infectious site, with splicing in the lung resembling DTT-treated cultures (iUPR) and splicing in the cornea being indistinguishable from control culture conditions (bUPR). To gain a deeper understanding of how the UPR might regulate adaptation under conditions of variable ER stress, we performed RNA-sequencing on wild-type and Δ*hacA* strains under three conditions: glucose minimal media (GMM) as a reference for the bUPR, GMM + dithiothreitol as reference for the iUPR, and gelatin minimal media that likely reflects the proteinaceous environment of the cornea. In the wild-type strain, gelatin media did not promote the upregulation of genes typically associated with the iUPR, suggesting nutrient limitation alone is not sufficient to acutely activate the canonical iUPR response. Nevertheless, the pathway does regulate key metabolic pathways that are unique to growth on gelatin, including those involved in alternative carbon utilization, cell wall homeostasis, redox balance, and secondary metabolism. These data suggest that the cornea fails to drive an iUPR response and instead leads to an intermediate level of UPR activation that has a unique transcriptional signature.

## Results

### The *A. fumigatus hacA* mRNA is not spliced beyond basal levels on proteinaceous substrates or in the infected cornea

In order to drive FK development, *A. fumigatus* must assimilate nutrients from and penetrate through the dense collagen matrix of the corneal stroma. Our interest in the fungal UPR as a potential virulence determinant in the cornea was therefore based on two simple and connected hypotheses: first, that *A. fumigatus* upregulates the expression of various secreted proteases *in vivo* and, second, that an upregulation in secretion would lead to ER stress and activate the iUPR as determined by increased *hacA* splicing above baseline. To confirm these predictions, we employed an epithelial abrasion model of FK developed and previously reported by our group [26,34]. Briefly, C57BL/6J mice were immunosuppressed with a single treatment of methylprednisolone on the day preceding inoculation. Although immunosuppression is not a prerequisite for the establishment of FK in our model, or indeed in patients, we have found that a single dose of steroids facilitates more uniform disease progression across animals. On the day of inoculation, the epithelium on the cornea was ulcerated and overlayed with *A. fumigatus* (Af293) conidia that were pre-germinated (swollen, but not polarized) in rich media. In contrast to the sham-inoculated controls, *A. fumigatus* corneas showed obvious signs of infection by 48 h post-inoculation (p.i.), including diffuse corneal opacification and surface irregularity (**Fig 1A**). Total RNA was isolated from infected corneas at this time point as well as from *A. fumigatus* cultured in glucose minimal medium (GMM), containing a preferred source of carbon (glucose) and nitrogen ($NH_4$), for 48 h as the baseline condition. Consistent with our first hypothesis, the cornea-derived samples displayed higher levels (50-1000-fold) of steady-state mRNA for each of *A. fumigatus* collagenase genes tested, including the metalloprotease (*mep1*—AFUA_1G07730), a serine alkaline protease (*alp1*—AFUA_4G11800), and two dipeptidyl-peptidases (*dppIV*—AFUA_4G09320 and *dppV*—AFUA_2G09030) (**Fig 1B**). These results support the interpretation that *A. fumigatus* upregulates its secreted hydrolytic activity to support invasive growth in the corneal environment.

Treatment of the *A. fumigatus* AfS28 background with dithiothreitol (DTT) leads to a rapid accumulation of misfolded proteins in the ER and the IreA-mediated splicing of the

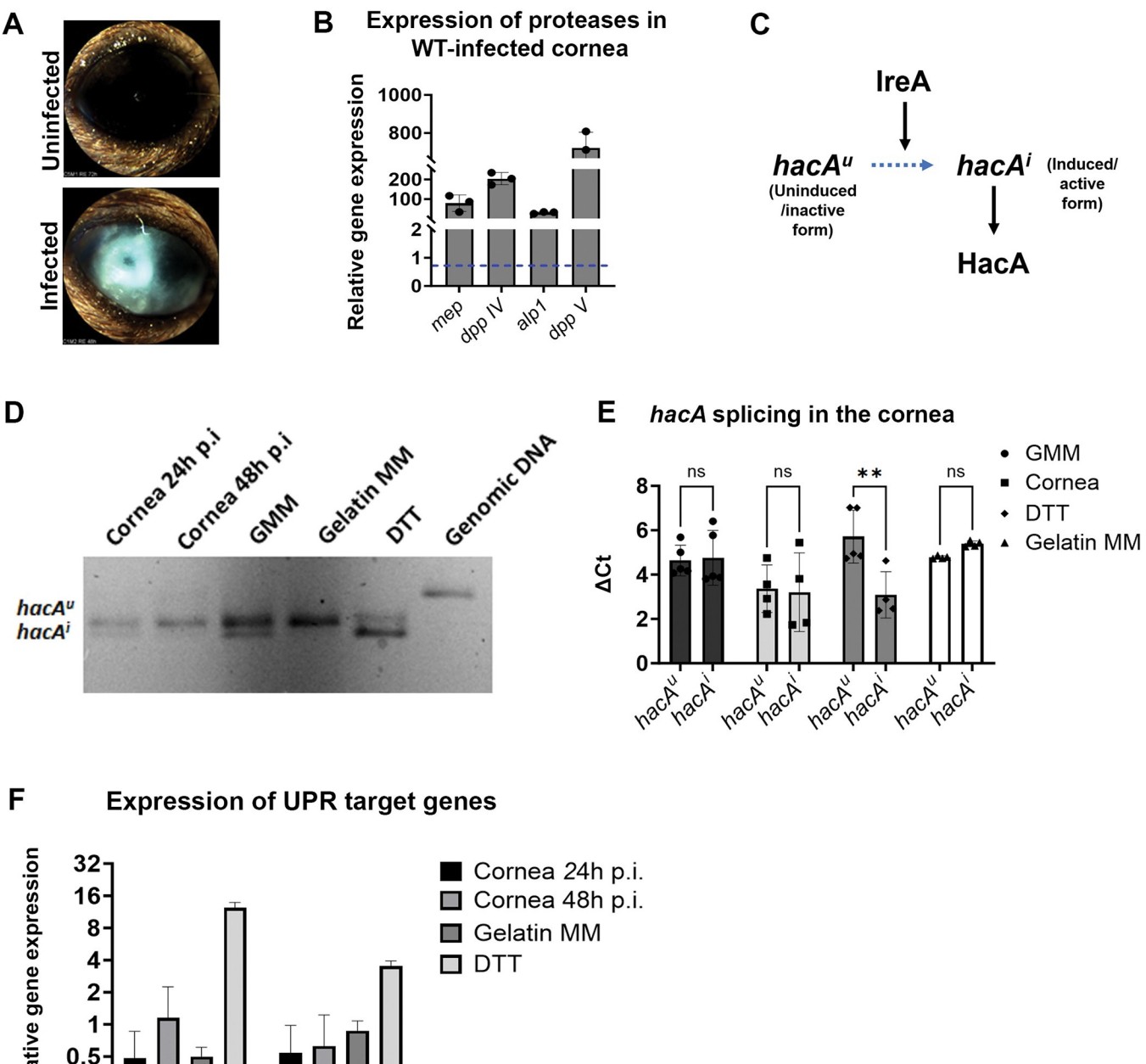

**Fig 1. *The A. fumigatus hacA mRNA is* not spliced beyond basal levels on proteinaceous substrates or in the infected cornea.** (A) Representative slit-lamp images revealing dense opacification in *A. fumigatus* (Af293)-infected corneas 48 h post-inoculation (p.i.). (B) qRT-PCR analysis *A. fumigatus* infected corneas (48 h p.i.) reveals an up-regulation of several fungal collagenase genes relative to GMM control cultures. β-tubulin was used as a reference gene and data represent the mean of three corneas. (C) Schematic diagram of *A. fumigatus* UPR, where the uninduced *hacA* splice form (*hacA^u*) is spliced by the IreA ribonuclease domain to form the induced (*hacA^i*) splice form which then encodes the HacA transcription factor. (D) The *hacA* mRNA was amplified by RT-PCR and the PCR products were separated on a 3% agarose gel to detect both the *hacA^u* (upper band) and *hacA^i* (lower band) forms in the *in vivo* (24 and 48 h p.i.) and the *in vitro* (GMM, gelatin MM, and DTT-treated) samples at 48 h. (E) The ΔCts of *hacA^u* to *hacA^i* mRNA normalized to actin. Data represent the mean of corneal samples isolated across two independent experiments where the cornea samples were taken at 48 h p.i., and (F) qRT-PCR of known UPR target genes–*bipA* and *pdiA* in the *A. fumigatus*-infected cornea at 24 h & 48 h p.i., gelatin MM and GMM treated with DTT, relative to the control (GMM).

*hacA* transcript from the uninduced (*hacA*u) to the induced/active (*hacA*i) form [15] (**Fig 1C**). We predicted that the increased expression of secreted proteases that occurs in the corneal environment, along with other stresses that are encountered during infection (e.g. oxidative stress from inflammatory cells), would similarly promote ER stress and a detectable accumulation of *hacA*i beyond the baseline condition, which, as in the above experiment, is *in vitro* cultivation on media (GMM) that does not require the breakdown by secreted hydrolases. To test this, we analyzed the relative abundance of *hacA* splice forms in *A. fumigatus* Af293 grown in the following conditions: 1) GMM, as the baseline (non-inducing) control; 2) GMM with 2 h DTT treatment as a positive control; 3) gelatin minimal medium (Gel MM), which contains gelatin (collagen hydrolysate) as the sole carbon and nitrogen source, and on which protease gene expression is induced similar to the cornea and; 4) 48 h murine corneas as described above. We performed RT-PCR using *hacA* primers that span the previously defined 20 bp non-conventional splice region, thus allowing for the detection of both the *hacA*u and *hacA*i forms following the resolution of the amplicons by gel electrophoresis. The samples were also analyzed by qRT-PCR using two primer sets that distinctly amplify the *hacA*u and *hacA*i cDNAs. Both forms of the *hacA* band were detected in the baseline sample, although the un-induced (higher molecular weight) band was more abundant. We interpret this steady state *hacA* splicing to correspond with the 'basal UPR' (bUPR) as previously defined by Feng and colleagues using *A. fumigatus* strain AfS28 [15,35]. The 2 h DTT treatment effectively reversed the relative intensity of the two bands, which is again consistent with previous reports and corresponds to the 'induced UPR' (iUPR). In contrast to our prediction, however, neither growth in gelatin nor the cornea shifted the splice form ratio beyond what was observed in GMM (**Fig 1D and 1E**). The *hacA* splicing data corresponded with the expression analysis of two known genes under the control of the HacA transcription factor, the protein chaperones *bipA* and *pdiA*, which were both induced in DTT but not in gelatin or the corneal samples (**Fig 1F**). In summary, these results suggest that only the bUPR of strain Af293 is active in the infected cornea as well as on culture media that reflects the proteinaceous composition of the cornea.

## Af293 HacA is required for the growth under conditions of acute ER stress and on protein-rich substrates

Given our above results, we reasoned that either 1) *A. fumigatus* bUPR activity is sufficient to support growth and virulence in the cornea, or 2) the UPR components are dispensable in this context. To explore these possibilities, we first sought to generate deletion mutants of the core UPR genes in Af293. To begin, the Af293 *hacA* coding sequence was replaced with the hygromycin resistance cassette (*hph*) using Cas9-mediated homologous recombination [36]. Subsequent generation of the complemented strain (*ΔhacA* C') was achieved through the ectopic integration of the WT *hacA* allele (fused with the *bleR* cassette) into the *ΔhacA* mutant (**S1A Fig**). Consistent with the AfS28 *ΔhacA*, and orthologous mutants in other fungi, the Af293 *ΔhacA* was hypersensitive to stressors that acutely misfold proteins (high temperature), inhibit the normal folding of nascent polypeptides in the ER (tunicamycin), or block the egress of proteins from the ER (brefeldin A) (**Figs 2A, 2B, and S1C**). Taken together, these results demonstrate a conserved essentiality of the UPR under conditions of acute ER stress in which the iUPR is functional. Also consistent with the AfS28 mutant, the Af293 *ΔhacA* displayed minor but obvious growth and conidiation defects under control/baseline conditions (YPD or GMM at 35˚C), which were fully restored to WT levels in the complemented strain (**Figs 2A, 2E, and S1B**). This supports the interpretation that bUPR buffers the endogenous ER stress associated with growth and developmental processes, even in otherwise favorable conditions.

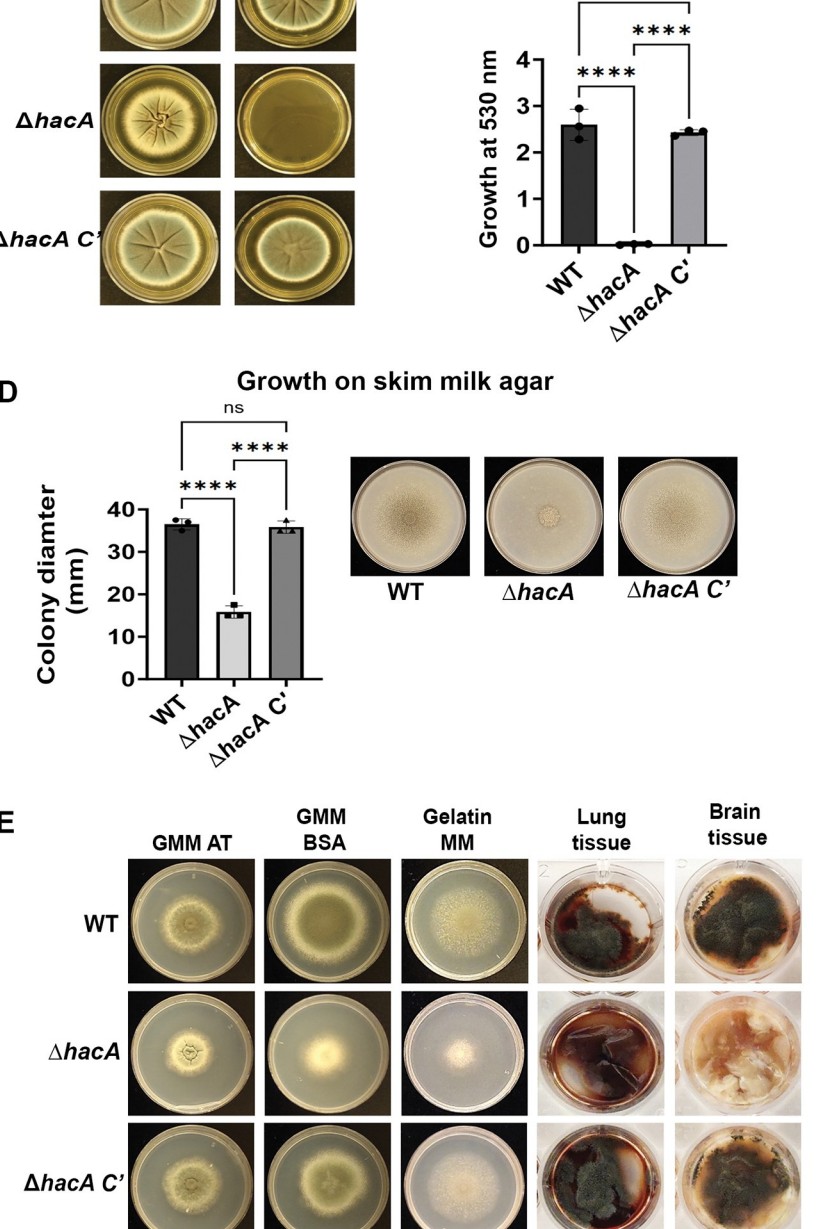

**Fig 2. Af293 HacA is required for the growth under conditions of acute ER stress and on protein-rich substrates.** (A) Representative images of the indicated strains grown on YPD plates for 72 h at 35˚C and 48˚C. (B) The strains were growth in static liquid GMM culture in the presence or absence of 6.25 μg/mL tunicamycin, an inhibitor of protein glycosylation; data represent the mean of triplicate samples analyzed by Brown-Forsythe and Welch's ANOVA **** <0.0001, ** 0.0022. (C) Culture supernatants were analyzed for azocollagen degradation [absorbance at 520nm normalized to dry weight (g)] after growth in GMM containing fetal bovine serum (GMM-FBS) for 72 h; data represent the mean of four replicate samples analyzed by Brown-Forsythe and Welch's ANOVA **** <0.0001, ** 0.0022. (D) Representative images of WT, *ΔhacA* and *ΔhacA C'* grown on 1% skim milk agar with growth depicted as colony diameter; data represented as a mean of triplicate samples analyzed by One-way Ordinary ANOVA **** <0.0001. (E) Representative images of the strains grown on media with containing ammonium tartrate (GMM-AT) or bovine serum albumin (GMM-BSA) as the sole nitrogen, gelatin as the sole carbon and nitrogen source (Gelatin MM), or murine lung and brain tissue, for 72 h at 35˚C.

To determine if the Af293 UPR has an enhanced role in supporting growth in proteina-ceous environments, which is what we predict is encountered in the collagen-rich cornea, we began by collecting fungal supernatants following growth in minimal media containing fetal bovine serum as the sole nitrogen source (GMM-FBS). In contrast to the WT and comple-ment-derived samples, *ΔhacA* supernatants displayed undetectable azocollagen degradation, suggesting that the secretion and/or expression of collagenases was impaired in the mutant (**Fig 2C**). Consistent with this, the *ΔhacA* mutant displayed a partial but significant growth defect on various protein-rich substrates (including GMM-BSA, gelatin minimal medium, and skim milk agar) relative to the WT and *ΔhacA* C' strains (**Fig 2D and 2E**). The zone of clear-ance around the *ΔhacA* colony on the skim milk plates indicated, however, residual protease activity that could at least partially account for the growth on these media. More strikingly, *ΔhacA* demonstrated almost unperceivable growth on mouse lung and brain tissues, suggest-ing an increased need for UPR activity on biological substrates (**Fig 2E**). In summary, these data demonstrate that although the UPR pathway is not demonstrably induced under condi-tions we suppose are nutrient limiting, (i.e. based on *hacA* splicing), it nevertheless plays a role in regulating the secretion of proteases and growth in such environments.

Interestingly, we were unable to isolate an *ireA* deletant across numerous transformation attempts utilizing the same CRISPR/Cas9 approach, as well as with standard homologous recombination techniques. As discussed, such a mutant was generated by Feng and colleagues in the AfS28 background and its growth and lung virulence phenotypes were overall more severe than the isogenic *ΔhacA* [15]. This suggests IreA retains functions beyond its role in splicing *hacA*, and we reasoned this functional importance may be enhanced, and conse-quently essential, in Af293. To evaluate this further, we next generated Af293 mutants in which the *ireA* or *hacA* native promoters were disrupted with the doxycycline/tetracycline-repressible promoter (tTA) using a Cas9-mediated recombination approach (**S2A** and **S2B Fig**) [37,38]. As expected, the WT Af293, $Ptet_{off}$-*hacA*, and $Ptetoff$-*ireA* strains were phenotypi-cally indistinguishable in the absence of doxycycline. Supplementation with doxycycline had no impact on the WT and mildly suppressed the growth and development of $Ptet_{off}$-*hacA* to levels that resembled the baseline phenotype of *ΔhacA* (**S2C Fig**). Strikingly, and by contrast, the $Ptet_{off}$-*ireA* mutant was severely growth ablated in the presence of doxycycline, irrespective of the nutrient content of the medium (**S2 Fig**). Taken together, these results support a model in which additional (*hacA*-independent) functions of IreA render it essential for the vegetative growth of Af293, even in low ER stress environments.

## The *A. fumigatus* UPR is essential for the establishment of corneal infection

We next wanted to determine if a basal level of *hacA* splicing supports *A. fumigatus* virulence in the cornea. Towards this end, the Af293 WT, *ΔhacA*, and *ΔhacA* C' strains were analyzed in the murine model described above. Remarkably, and in contrast to the progressive disease development that occurred with the WT and complement infected corneas, animals inoculated with *ΔhacA* failed to demonstrate any signs of disease on external evaluation and disease scor-ing criteria (**Fig 3A and 3B**). Alterations in the corneal structure were evaluated more thor-oughly with optical coherence tomography (OCT), which provides a cross-sectional image of the anterior segment in live animals for downstream and morphometric analysis. OCT revealed thickened corneal tissue and alterations in refraction for WT and complement-infected animals, indicating the presence of edema and inflammation that are characteristic of FK (**Fig 3C**). By contrast, and consistent with the external images, corneas inoculated with the *ΔhacA* mutant were indistinguishable from the sham-inoculated controls (**Fig 3D**). Tissue sec-tions taken 72 h p.i. were consistent with the OCT findings, demonstrating that WT and

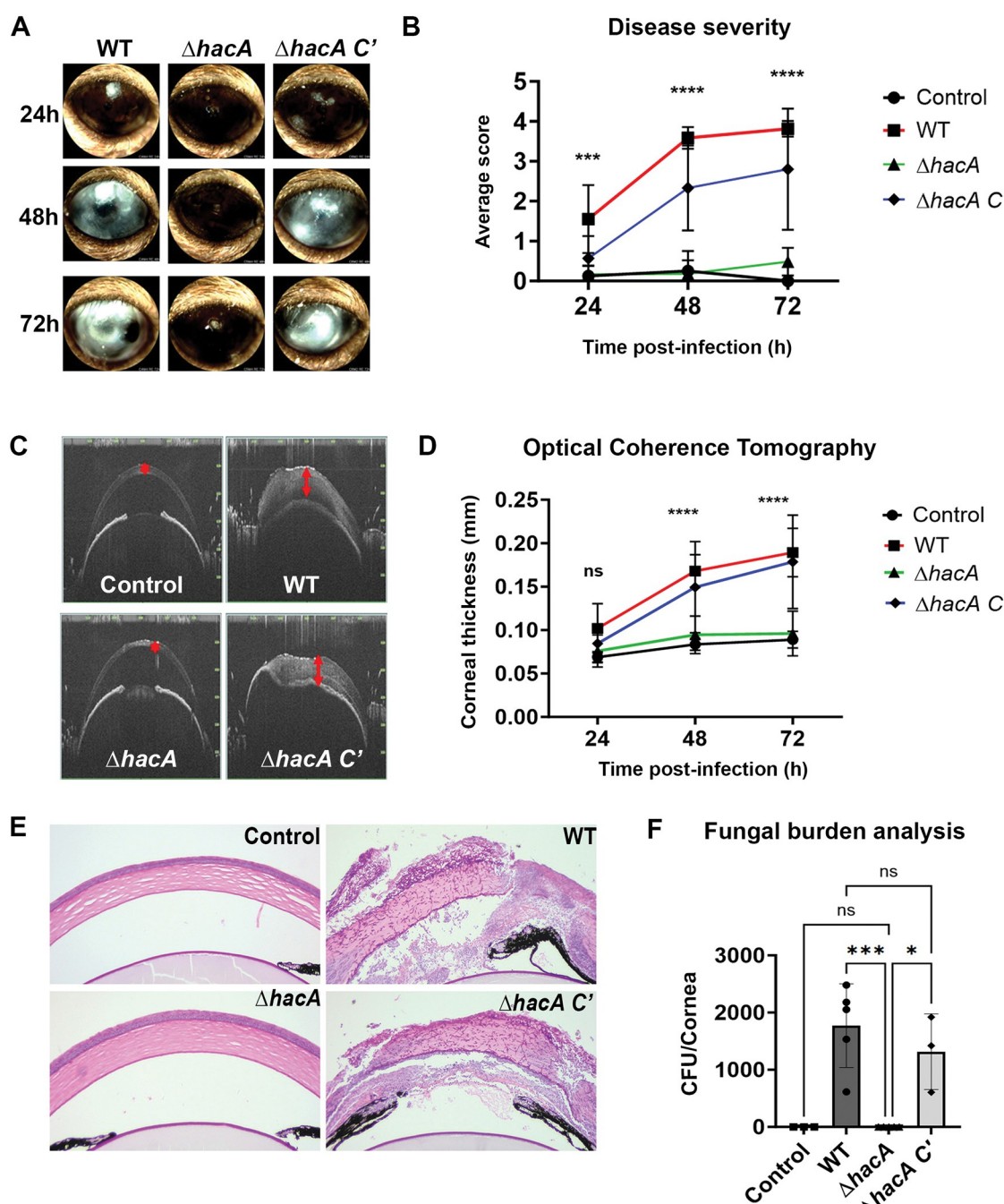

**Fig 3. The *A. fumigatus* UPR is essential for the establishment of corneal infection.** (A) Representative slit-lamp images of infected murine corneas at 24, 48, and 72 h p.i., and (B) average clinical scores (n = 10/group) over the course of the infection; data analyzed by Ordinary two-way ANOVA, p-value **** <0.0001 (C) Representative cross-sectional images of the cornea from OCT 72 h p.i.. (D) Corneal thickness measured across the group (n = 10/group) over the course of the infection averaging 13 points across the cornea; data analyzed by Ordinary two-way ANOVA p-value **** <0.0001. (E) Sections of cornea 72 h p.i. were stained with Periodic Acid Schiff and hematoxylin (PASH) to observe hyphal growth (magenta) and inflammatory immune cells (purple). (F) Fungal burden at 72 h p.i. as determined by colony forming unit (CFU) analysis from corneal homogenates; data analyzed by Ordinary one-way ANOVA p-value *** 0.0006.

complement infected corneas were marked by ulcerated and structurally abnormal corneas, with massive immune cell infiltration in both the stroma and anterior chamber. Fungal hyphae were also observed through the depth of the cornea in these two groups on histology, and homogenized and plated corneas from the same time point indicated similar fungal loads (**Fig 3E and 3F**). Histology of the *ΔhacA* corneas, by contrast, displayed normal epithelial and stromal architecture, minimal cellular infiltration, and no visible fungal growth which was confirmed in the CFU analysis (**Fig 3E**). These results, which suggest a critical role for the UPR in the establishment of corneal infection, were replicated with the full isogenic set of strains in an independent experiment using both male and female animals (**S3 Fig**).

## A loss in *A. fumigatus* secreted protease activity does not result in reduced virulence in the cornea

We next hypothesized that the avirulent phenotype of *ΔhacA* in the cornea was largely attributable to its loss in extracellular collagenase activity, which we predicted would be required for growth through the stromal ECM. If true, we reasoned that an *A. fumigatus* strain that is deficient in protease activity, but otherwise retains a functional UPR, would similarly be attenuated for virulence in the cornea. We therefore generated deletion mutants of *prtT* and *xprG*, which encode distinct transcription factors that control a large and partially overlapping subset of protease-encoding genes in *A. fumigatus* [39,40]. The Af293 genome harbors a single *prtT* ortholog (AFUA_4G10120), which we replaced with a *bleR* cassette by split marker transformation (**S4A Fig**). Consistent with what was reported previously, our *ΔprtT* mutant was indistinguishable from WT on rich media, but markedly growth and developmentally impaired on media containing gelatin or BSA as the sole carbon or nitrogen source (**Fig 4A**). Interestingly, the Af293 genome harbors two putative *xprG* orthologs with almost near sequence identity (96.2%) across the protein-coding and flanking regions, which we predict is due to a recent gene duplication event in this background. While the first locus (AFUA_8G04050) is predicted to encode a fully functioning transcription factor, the second (AFUA_1G00580) contains two frame shifts within the coding region that result in a partial loss of the putative DNA binding domain (**S4D Fig**); we refer to these genes as *xprG1* and *xprG2*, respectively. Due to the high sequence similarity across these loci, it was not feasible to design guide RNAs that were specific to either gene. So, while the transformation of cross-reactive Cas9 ribonucleoproteins (RNPs) into the Af293 WT background resulted in the isolation of *xprG2* single and *xprG1/xprG2* double deletants, we did not isolate a single *xprG1* mutant. Consistent with the prediction that *xprG2* encodes a non-functional protein, the *ΔxprG2* strain displayed indistinguishable growth and developmental phenotypes from WT on both rich and proteinaceous media (**Fig 4A**). The *ΔxprG1ΔxprG2*, on the other hand, was phenotypically similar to the *ΔprtT* mutant, suggesting that XprG1 is a functional regulator of protease activity. Transformation of the *ΔprtT* mutant with the *xprG1/2* cross-reactive RNPs resulted in the isolation of triple mutants (*ΔprtTΔxprG1ΔxprG2*), which did not display an obvious enhanced growth defect on proteinaceous media beyond that of *ΔprtT* or *ΔxprG1ΔxprG2* (**Fig 4A**). The azocoll degradation assay revealed equivalent levels of extracellular collagenolytic activity between WT and *ΔxprG2* strains, as expected, but undetectable levels of activity in the *ΔprtT*, *ΔxprG1ΔxprG2*, and *ΔprtTΔxprG1ΔxprG2* mutants (**Figs 4B** and **S4C**). Notably, there was no difference in the level of detectable azocoll degradation between these protease-deficient mutants and *ΔhacA*.

We next tested the virulence of the Af293 WT, *ΔprtT*, and *ΔprtTΔxprG1ΔxprG2* strains in our model of FK. Interestingly, and contrary to our hypothesis, both of the protease mutants displayed comparable virulence to the WT with respect to clinical appearance, corneal tissue thickness, pathology, and fungal burden (**Fig 4C–4G**). Thus, it appears that proteases under the control of

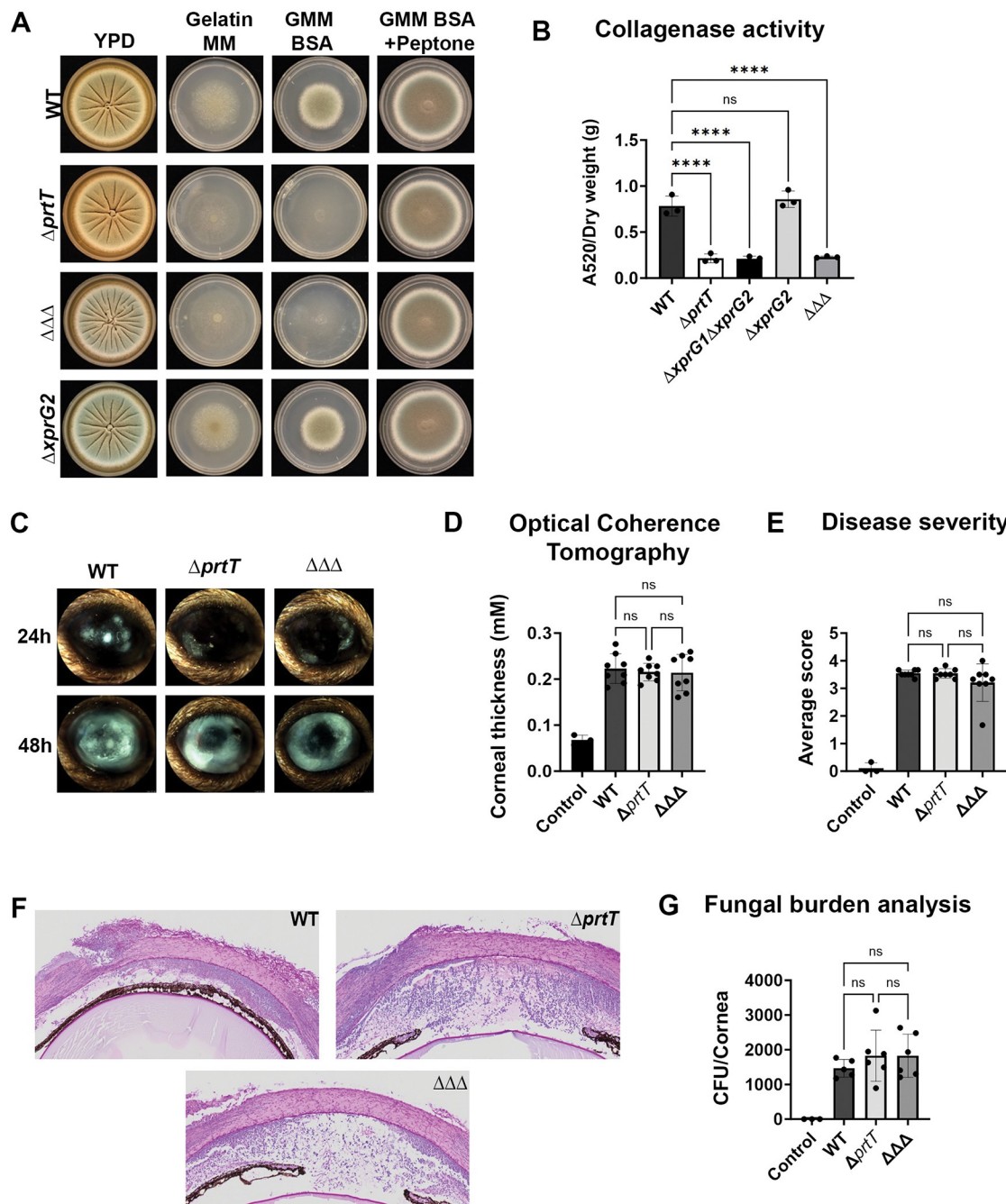

**Fig 4. A loss in *A. fumigatus* secreted protease activity does not result in reduced virulence in the cornea.** (A) Conidia from WT, *ΔprtT*, *ΔxprG2*, *ΔprtTΔxprG1ΔxprG2* (ΔΔΔ) were inoculated onto YPD, gelatin MM, GMM BSA and GMM BSA+peptone and incubated at 35°C for 72 h. (B) Culture supernatants were analyzed for azocollagen degradation [absorbance at 520 nm normalized to dry weight (g)] after growth in GMM-FBS for 72 h at 35°C at 200 rpm; data represent the mean of three replicate samples analyzed by Ordinary one-way ANOVA **** <0.0001. (C) Representative slit-lamp images of infected murine corneas at 24 h and 48 h p.i. (D) Corneal thickness measured across the group (n = 8/group) 48 h p.i. averaging 13 points across the cornea; data analyzed by Ordinary one-way ANOVA. (E) Average clinical scores (n = 8/group) 48 h p.i.; Data analyzed by Ordinary one-way ANOVA. (F) Sections of cornea 48 h p.i. were stained with Periodic Acid Schiff and hematoxylin (PASH) and reveal fungal growth and inflammation in all samples. (G) Tissue fungal burden as assessed by CFU analysis 48 h p.i.; data analyzed by Ordinary one-way ANOVA.

PrtT and/or XprG1 are not required for the growth of *A. fumigatus* in the cornea, and we cannot conclude that the avirulence phenotype of *ΔhacA* is attributable to a loss in proteolytic activity.

## *A. fumigatus hacA* is spliced and essential for virulence in a murine model of invasive pulmonary aspergillosis

As mentioned, previous studies have shown the AfS28 *ΔhacA* mutant is attenuated for virulence in various murine models of IPA. Notably, however, *ΔhacA*-infected lungs still display moderate fungal growth and inflammation and ~50% of the animals succumb to infection [14,15]. This contrasts our current FK results, in which none of the corneas inoculated which Af293 *ΔhacA* develop outward signs of disease or retain viable fungus. Thus, we wanted to determine if our mutant is similarly avirulent in an IPA model (which would indicate differences between the AfS28 and Af293 backgrounds), or if the pathogenic potential of *ΔhacA* does indeed differ between the lung and corneal environments. Towards this end, and in an attempt to keep our lung and corneal models as similar as possible, we began by inoculating C57BL/6J male mice with WT Af293, via intranasal instillation, following immunosuppression with methylprednisolone. None of the inoculated animals developed signs of infection, so we turned to a well-established IPA model in which male outbred (CD-1) mice were immunosuppressed with multiple doses of triamcinolone acetonide [41–43]. In this setting, we observed high cumulative mortality for the WT and *ΔhacA C'* infected groups, where histological examination revealed the presence of progressively larger fungal/inflammatory lesions over time (Fig **5A** and **5B**). By contrast, the cumulative mortality of the *ΔhacA*-infected animals was statistically indistinguishable from the uninfected controls. Interestingly, sporadic and small areas of fungal growth were observed *ΔhacA* lungs up to 14 days p.i., suggesting the mutant retains the capacity to germinate and persist in the pulmonary environment, albeit to levels insufficient to drive disease development. We then ran our FK infection experiment in this model (CD-1 male mice immunosuppressed with triamcinolone) and, consistent with the previous experiments, found no signs of disease or fungal persistence in *ΔhacA*-inoculated corneas. A large proportion of WT-infected animals also failed to develop infection in this model, however, suggesting outbred mice are inherently resistant to Af293 infection in the cornea, despite being susceptible in IPA. Nevertheless, these data cumulatively suggest that Af293 *ΔhacA* is avirulent in both the lung and corneal environments, although the capacity of the mutant to persist, or the kinetics through which it is cleared, may differ between the two.

We next wanted to determine if the basal level of *hacA* splicing observed in the cornea was also supporting *A. fumigatus* virulence in the lung. Accordingly, total RNA was extracted from CD-1 mouse lungs at various days post-inoculation with WT Af293. The resulting cDNA was then analyzed by endpoint and qPCR to assess the ratio of $hacA^u$ and $hacA^i$ splice forms as described above. RNA was isolated in parallel from DTT-treated and untreated GMM cultures to serve as positive and negative control references, respectively. Interestingly, all of the lung-derived samples displayed $hacA^i/hacA^u$ ratios that were comparable to the DTT-treated samples, indicating that the *A. fumigatus* iUPR is operative in this tissue (**Fig 5C**). Thus, although HacA is required for virulence in both the cornea and lung, the amount of ER stress experienced during infection, and hence the degree of UPR activation, appears to vary across these host tissues.

## Transcriptomics supports a distinct role for the UPR across environments of variable ER or nutritional stress

The above data cumulatively suggest that the *A. fumigatus* UPR promotes adaptation to a variety of environments that vary in the amount of ER stress they impart to the fungus. What remained unclear, however, was whether the UPR exists is a simple binary switch (i.e. bUPR

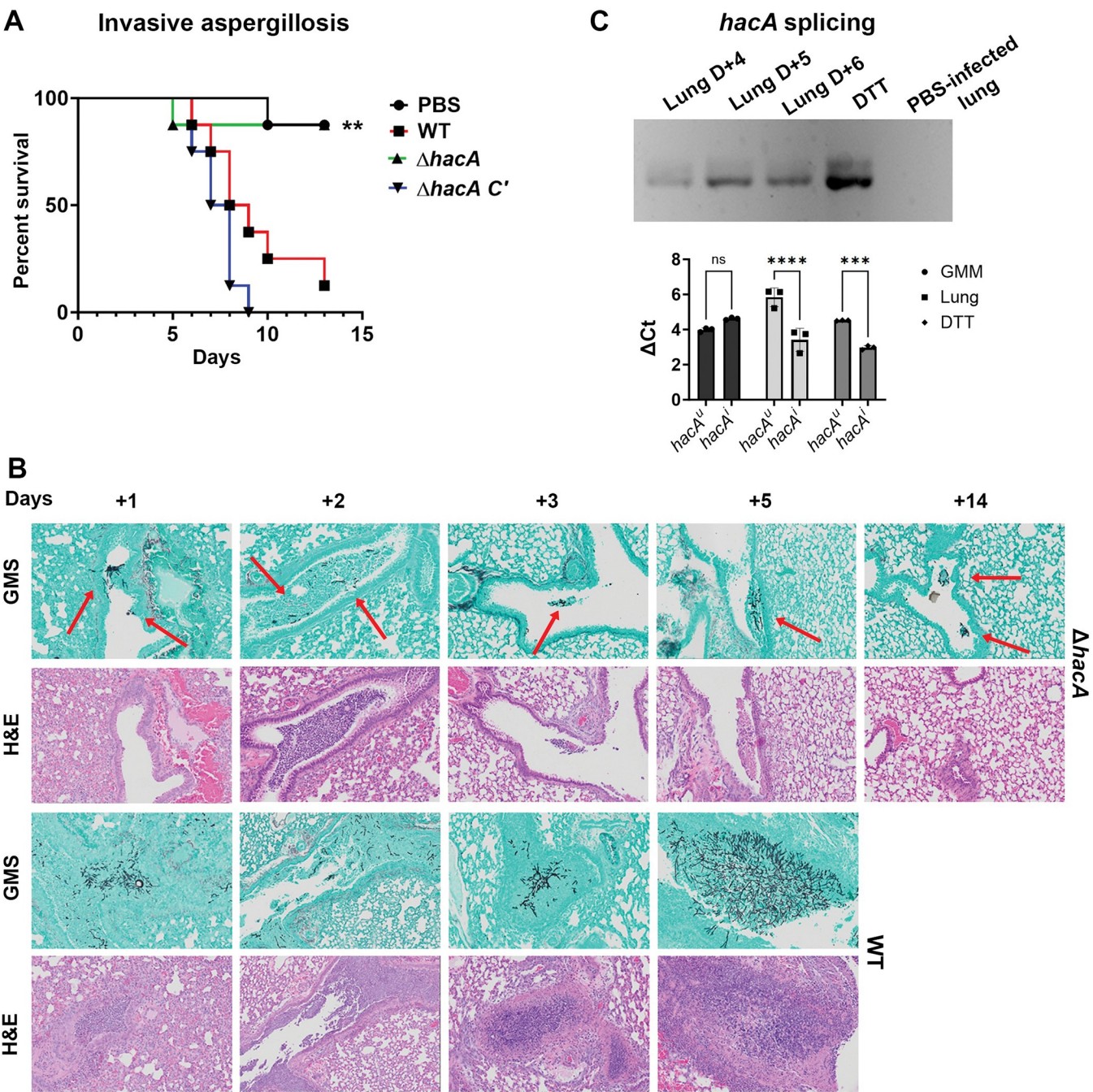

**Fig 5. *A. fumigatus hacA* is spliced and essential for virulence in a model of invasive pulmonary aspergillosis.** (A) Cumulative mortality of triamcinolone immunosuppressed CD-1 mice inoculated with 2 x 10$^6$ conidia via intranasal instillation (n = 8/group, n = 5 for PBS group); survival curves were compared using the logrank Mantel–Cox test where *ΔhacA* vs WT p-value = 0.0067, *ΔhacA* vs *ΔhacA C'* p-value = 0.0015. (B) Histological staining showing fungal growth, stained black by GMS, and inflammatory cells, stained purple by H&E. Lungs inoculated with *ΔhacA* demonstrate limited germination starting on D+1 up to D+14 (highlighted with red arrows). (C) *Top panel*: RT-PCR *hacA* amplicons separated on a 3% agarose gel to detect both the *hacA$^u$* and *hacA$^i$* forms in the WT-infected lungs along with the DTT-treated cultures (positive splicing control) and sham-inoculated lungs (negative control) from 48 h. *Bottom panel*: ΔCts of *hacA$^i$* and *hacA$^u$* products as determined by qRT-PCR using splice form specific primers sets; data represented as mean of triplicate samples analyzed by two-way ANOVA **** <0.0001 *** 0.0005.

vs. iUPR), or if the HacA readout was more finely tuned to the environment. Accordingly, we sought to characterize the HacA-dependent transcriptome under three conditions that represent various levels of ER stress: 1) GMM, as a "low" ER stress condition with basal levels of *hacA* splicing (bUPR); 2) GMM + DTT, as a "high" ER stress condition that more closely mirrors the lung in regard to $hacA^i/hacA^u$ ratios (iUPR), and 3) Gelatin MM, as a putatively "intermediate" ER stress condition that represents the cornea in terms of its proteinaceous nutrient content and in terms of the level *hacA* splicing observed. Differentially expressed genes (DEGs) for WT were defined as those with a 4-fold change relative to the GMM condition; DEGs for *ΔhacA* were those with a 4-fold change relative to the WT in that particular condition. A full list of the DEGs across all treatments, as well as a complete list of enriched FunCat and GOterms can be found in the **S1–S5 Data files**.

**GMM.**   Despite serving as a nominally homeostatic condition, 514 genes were downregulated in *ΔhacA* (relative to WT) in GMM, including numerous chaperones that fit with the known function of the UPR (e.g. *bipA*, *hsp78*, *hsp98*, *hsp30*) (**Fig 6**). Also downregulated in *ΔhacA* were numerous genes encoding proteins destined for the plasma membrane or secretion, including nutrient and drug transporters, proteases, and polysaccharide binding and degrading enzymes. Beyond the breakdown of polysaccharides as a nutrient source, it is likely

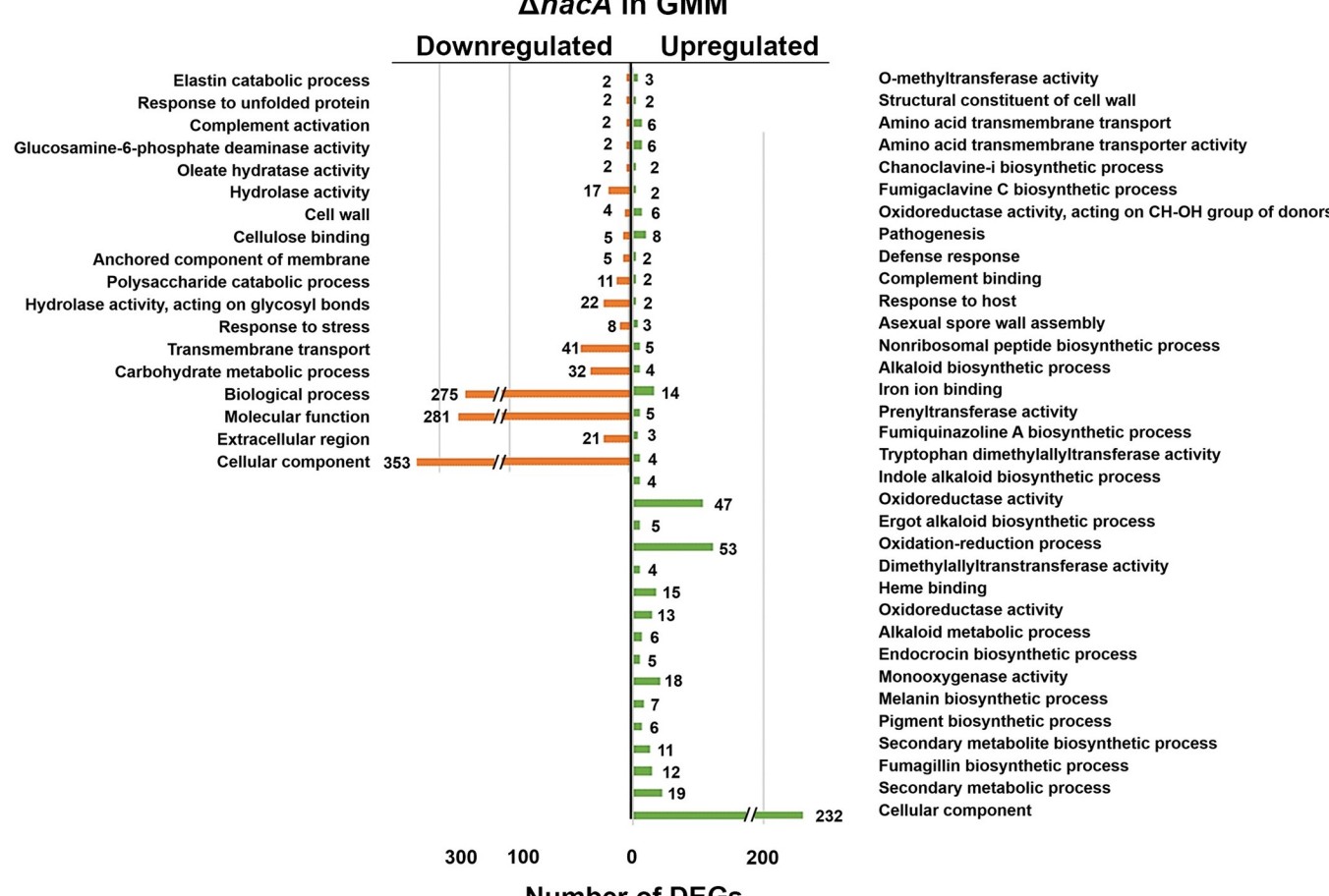

**Fig 6. Term-enrichment analysis of differently expressed genes in *ΔhacA* versus WT in glucose minimal medium.** Significantly enriched Gene Ontology (GO) terms among the *ΔhacA* DEGs in GMM, based on a 4-fold difference in abundance relative to WT. The numbers next to the bars reflect the total number of genes in the respective category.

many of such genes are involved in the routine maintenance of cell wall (e.g. glucanases, chitinases), and this corresponded to hypersensitivity of *ΔhacA* to cell wall stressors such as Congo red (**S5G Fig**). Thus, during growth in GMM, basal HacA function appears to not only support the expression of genes that regulate ER homeostasis and secretory function, but also broadly regulates the expression of genes that traffic through the secretory pathway. The dysregulation of these genes in *ΔhacA* likely accounts for the growth and developmental defects observed on GMM or rich media (**Fig 2**). Interestingly, 328 genes were upregulated in *ΔhacA* relative to the WT, and these were enriched in various secondary metabolic pathways (fumagillin, fumigaclavine C, fumiquinazoline), iron and heme binding, and oxidoreductase activity (**Fig 4B**). Although we predict that the expression of these pathways would not impact growth on GMM per se, their de-repression may be energetically unfavorable and further contribute to the growth defects of *ΔhacA* as well as altered sensitivities to the iron chelator BPS (**S5F Fig**).

**DTT.** DTT treatment of the WT led to the induction of 345 genes (relative to untreated GMM), and included an enrichment of those within the 'protein folding and stabilization' Functional Category (FunCat). Such genes – including protein chaperones (*bipA*, *hsp70*), calnexin, protein isomerases, and ER resident oxidoreductases and ATPases – were similarly found to be induced in the *A. fumigatus* AfS28 background by microarray, and therefore appear to constitute a conserved iUPR [14]. Other enriched FunCat terms in our DTT-induced gene set included 'detoxification by modification', 'secondary metabolism', 'oxidative stress', and 'extracellular polysaccharide degradation' (**Table 1**). 141 genes were down-regulated in DTT, many of which had overlapping enrichment in iron and oxido-reduction GO terms, or in secondary metabolism categories, including fumagillin biosynthesis. In summary, DTT treatment resulted in the expected induction of canonical ER stress-related genes; however, it is currently unclear if the altered expression of genes in a myriad of primary or secondary metabolic processes is directly tied to ER stress and UPR signaling.

Relative to the WT in DTT, 314 genes were downregulated in *ΔhacA*. Not surprisingly, enriched among these DEGs included those in the 'response to unfolded protein', 'endoplasmic reticulum', and 'protein glycosylation' GO terms, which altogether confirm the role of HacA in the primary transcriptional response that restores protein folding homeostasis to the ER (**Fig 7**). Interestingly, nearly as many genes (273) were upregulated in *ΔhacA* relative to WT, and these DEGs were enriched among secondary metabolic processes, including the 'fumagillin' biosynthesis' category that was down-regulated in WT upon DTT treatment. This suggests that HacA plays a role in the repression of several genes, either directly or by positively regulating a downstream repressor.

**Gel MM.** Growth of the WT in gelatin medium led to the upregulation of 672 genes (relative to GMM) with term enrichments that indicated a broad shift towards the metabolism of protein and other polymeric substrates, such as fatty acids and polysaccharides (**Table 1**). Despite this inferred dependency on increased secretory activity, we did not observe a corresponding increase in genes involved in protein folding or ER homeostasis like there was in DTT. Indeed, only 110 genes overlapped between the DTT and Gel MM induced gene sets in WT, and these overlapping genes were enriched in two categories: 'secondary metabolism' and polysaccharide binding'. Interestingly, the sensitivity of Af293 to brefeldin A, which induces ER stress by blocking vesicular transport through the Golgi, was increased on gelatin compared to GMM (**S1C Fig**). Taken together with the *hacA* splicing results, it appears that the fungus is under some degree of ER stress in protein-rich environments, just not to the threshold required for iUPR activation. In addition to the upregulation of secreted hydrolase genes, we further observed an upregulation of genes enriched for amino acid transport, gluconeogenesis, oxidative metabolism (electron transport), and oxidation-reduction, all of which seemingly corresponded to a broader metabolic shift to alternative carbon (non-glucose) utilization

**Table 1. Statistically altered gene categories (FunCat terms) for the WT Af293 following growth in gelatin or upon treatment with DTT; both relative to the GMM reference condition.**

| | FunCat ID | FunCat description | Adjusted p-value | # genes / input |
|---|---|---|---|---|
| Gelatin MM | 1.2 | Secondary metabolism | 2.877882e-17 | 122 / 672 |
| | 20.09.18.07 | Non-vesicular cellular import | 1.14E-05 | 33 / 672 |
| | 1.05 | C-compound and carbohydrate metabolism | 2.75E-05 | 90 / 672 |
| | 2.25 | Oxidation of fatty acids | 2.75E-05 | 16 / 672 |
| | 16.21.01 | Heme binding | 9.47E-05 | 16 / 672 |
| | 01.06.05 | Fatty acid metabolism | 0.000106 | 25 / 672 |
| | 01.01.11.04.02 | Degradation of leucine | 0.000106 | 9 / 672 |
| | 20.01.03 | C-compound and carbohydrate transport | 0.000124 | 34 / 672 |
| | 01.01.09.04.02 | Degradation of phenylalanine | 0.00022 | 11 / 672 |
| | 20.03 | Transport facilities | 0.000244 | 52 / 672 |
| | 20.09.18 | Cellular import | 0.000247 | 36 / 672 |
| | 16.21.15 | Biotin binding | 0.000293 | 6 / 672 |
| | 1.06 | Lipid, fatty acid and isoprenoid metabolism | 0.000701 | 61 / 672 |
| | 20.01.23 | Allantoin and allantoate transport | 0.00138 | 12 / 672 |
| | 01.01.09.05.02 | Degradation of tyrosine | 0.001494 | 7 / 672 |
| | 01.05.03 | Polysaccharide metabolism | 0.003192 | 30 / 672 |
| | 2.09 | Anaplerotic reactions | 0.005012 | 3 / 672 |
| | 20.01.15 | Electron transport | 0.01037 | 34 / 672 |
| | 32.07.01 | Detoxification involving cytochrome P450 | 0.023295 | 10 / 672 |
| | 16.21.05 | FAD/FMN binding | 0.028143 | 16 / 672 |
| | 32.07.07.01 | Catalase reaction | 0.036851 | 3 / 672 |
| | 01.20.33 | Metabolism of secondary products derived from L-tryptophan | 0.037059 | 5 / 672 |
| DTT | 1.2 | Secondary metabolism | 6.331832e-8 | 50 / 345 |
| | 32.07.03 | Detoxification by modification | 0.005024332 | 7 / 345 |
| | 32.05.01 | Resistance proteins | 0.01219727 | 9 / 345 |
| | 32.01.01 | Oxidative stress response | 0.02112102 | 9 / 345 |
| | 14.01 | Protein folding and stabilization | 0.04406497 | 10 / 345 |
| | 01.25.01 | Extracellular polysaccharide degradation | 0.04406497 | 5 / 345 |

(**Table 1**). Surprisingly, however, we also observed an upregulation of genes involved in iron/ heme binding as well as several secondary metabolic clusters, including gliotoxin. While these genes likely do not promote growth in gelatin MM directly, they may be broadly de-repressed as a part of the adaptive response within nutrient-limiting environments, e.g. through carbon catabolite de-repression, that may be concomitantly stressful in other ways. However, there were 493 down-regulated genes in Gel MM (relative to GMM), and these were similarly enriched in a variety of oxidoreductive and secondary metabolic processes. Therefore, while we can conclude there are broad transcriptional differences in *A. fumigatus* in glucose- versus proteinaceous media, it is difficult to reconcile why many of the specific pathways differ between the two.

311 genes were downregulated in *ΔhacA* (relative to WT) in the gelatin media. Whereas several of the enriched Gene Ontology (GO) terms for these DEGs were similarly observed among the GMM (e.g., hydrolases, polysaccharide binding/cell wall, hsp/chaperones) DEG list, the majority were unique to gelatin MM, including oxidation-reduction, iron binding, secondary metabolism, and pigment/melanin biosynthesis categories (**Fig 8**). Many of these gene categories were upregulated in the WT Af293 in gelatin (relative to WT in GMM), suggesting that HacA plays an important role in regulating the transcriptional adaptation of *A. fumigatus*

### Δ*hacA* in DTT

**Fig 7. Term-enrichment analysis of differently expressed genes in *ΔhacA* versus WT following DTT treatment.** Significantly enriched Gene Ontology (GO) terms among the *ΔhacA* DEGs in GMM + 2h DTT, based on a 4-fold difference in abundance relative to WT. The numbers next to the bars reflect the total number of genes in the respective category.

to the nutrient environment. The dysregulation of several genes from the RNA-seq dataset was validated in a replicated experiment using qRT-PCR (**Fig 8B**). 89 genes were upregulated in *ΔhacA* relative to WT, and these were enriched in just a few FunCat terms, including peptide transport and extracellular polysaccharide degradation.

Taken together, our transcriptomics analysis revealed a unique HacA-dependent transcriptome across three distinct conditions. Indeed, only 16 genes were common down-regulated *ΔhacA* DEGs across the three conditions, and only 7 were common up-regulated DEGs (**Fig 9 and S6 and S7 Data files**). In all cases, however, HacA appears to regulate not only genes involved in ER and secretory homeostasis, but also those involved in broader cellular shifts in primary and secondary metabolism, mitochondrial function and redox balance, and cell wall homeostasis.

## Discussion

Our current understanding of fungal UPR regulation and downstream signaling is largely based on studies that employ chemical stressors that are acutely toxic to the ER, induce detectable levels of *hacA/hac1* mRNA splicing, and markedly upregulate a conserved suite of genes (e.g. chaperones) that address an otherwise lethal level of stress. While these experimental parameters do indeed highlight the remarkable homeostatic capacity of the UPR, it remains unclear if these laboratory conditions reflect the true state of the pathway during saprophytic and pathogenic growth. In this regard, a key finding of our study is that the UPR-dependent transcriptome of *A. fumigatus* is distinct on glucose versus protein-rich media, with only ~10% of all *ΔhacA* DEGs being shared, despite the *hacA$^u$/hacA$^i$* ratios being indistinguishable

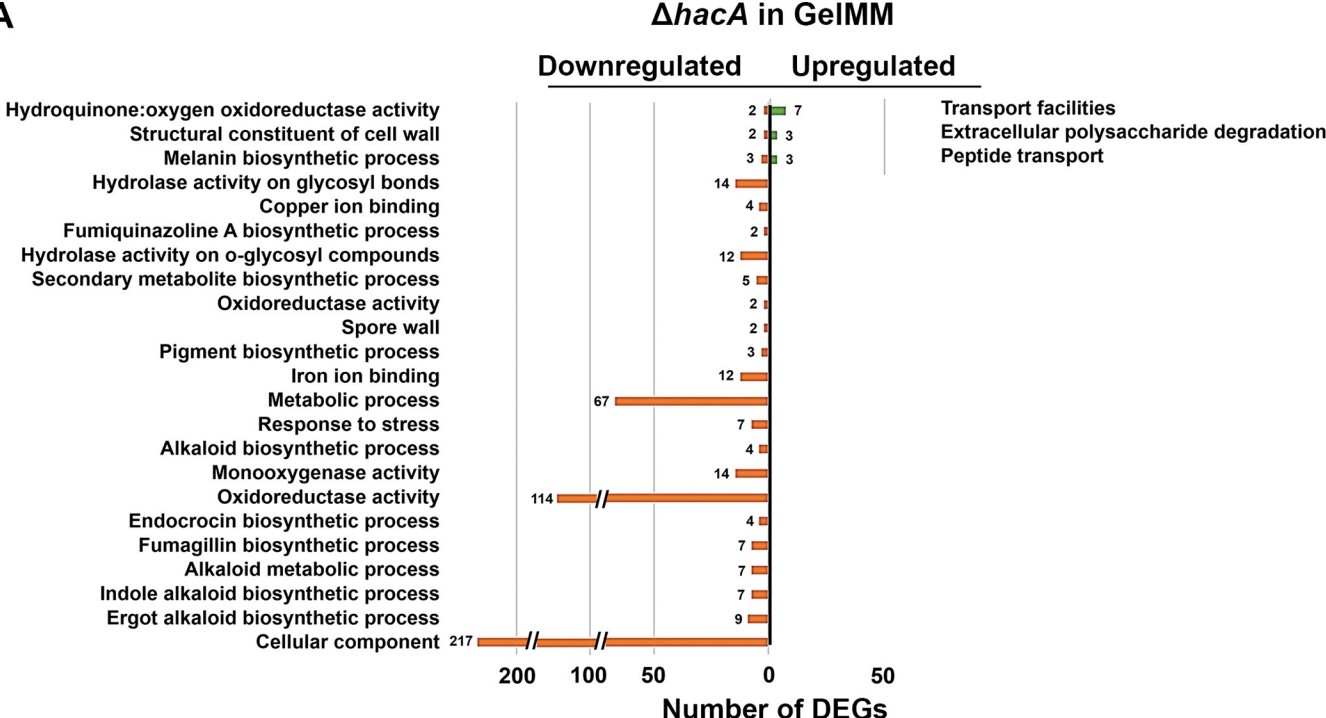

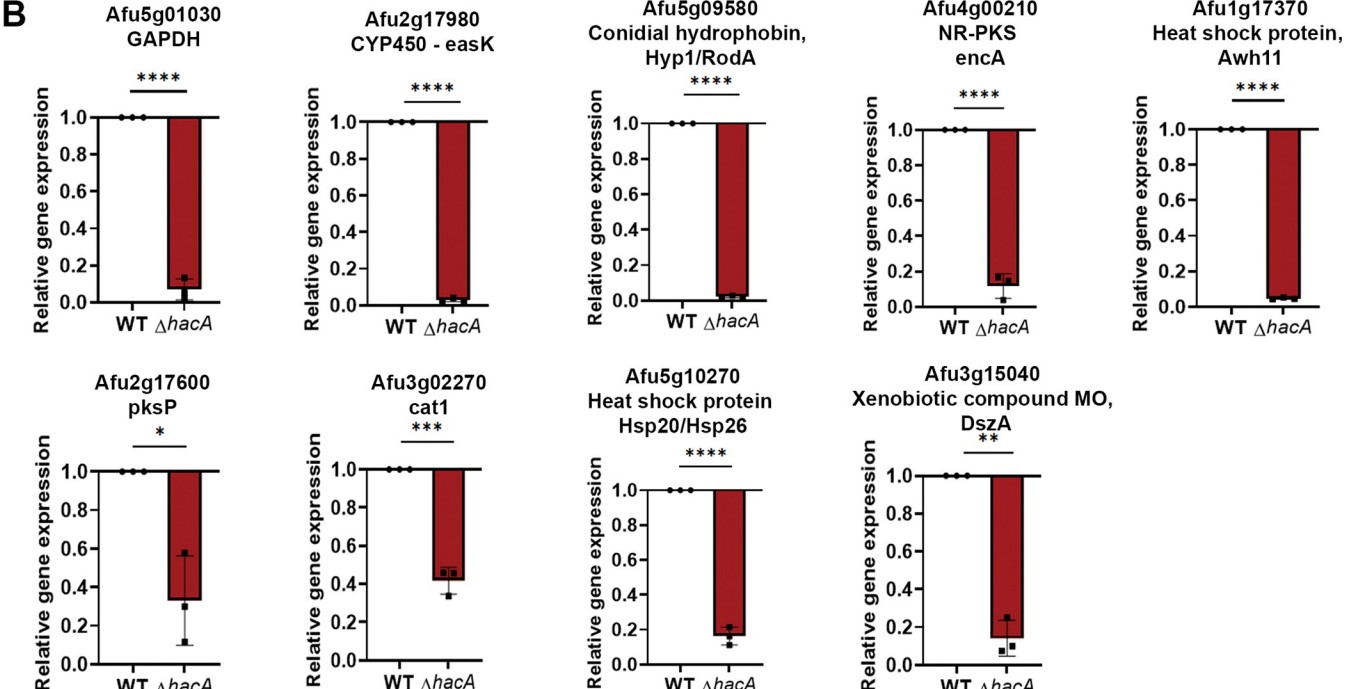

**Fig 8. Term-enrichment analysis of differently expressed genes in *ΔhacA* versus WT in gelatin minimal medium.** Significantly enriched Gene Ontology (GO) and FunCat terms among the *ΔhacA* DEGs in Gel MM, based on a 4-fold difference in abundance relative to WT. The numbers next to the bars reflect the total number of genes in the respective category. (B) qRT-PCR validation of selected *ΔhacA* DEGs from an independent experiment. Actin was used as the reference gene and data represent a mean of triplicate samples analyzed using the unpaired T-test ** 0.0076, *** 0.0001, **** 771 <0.0001.

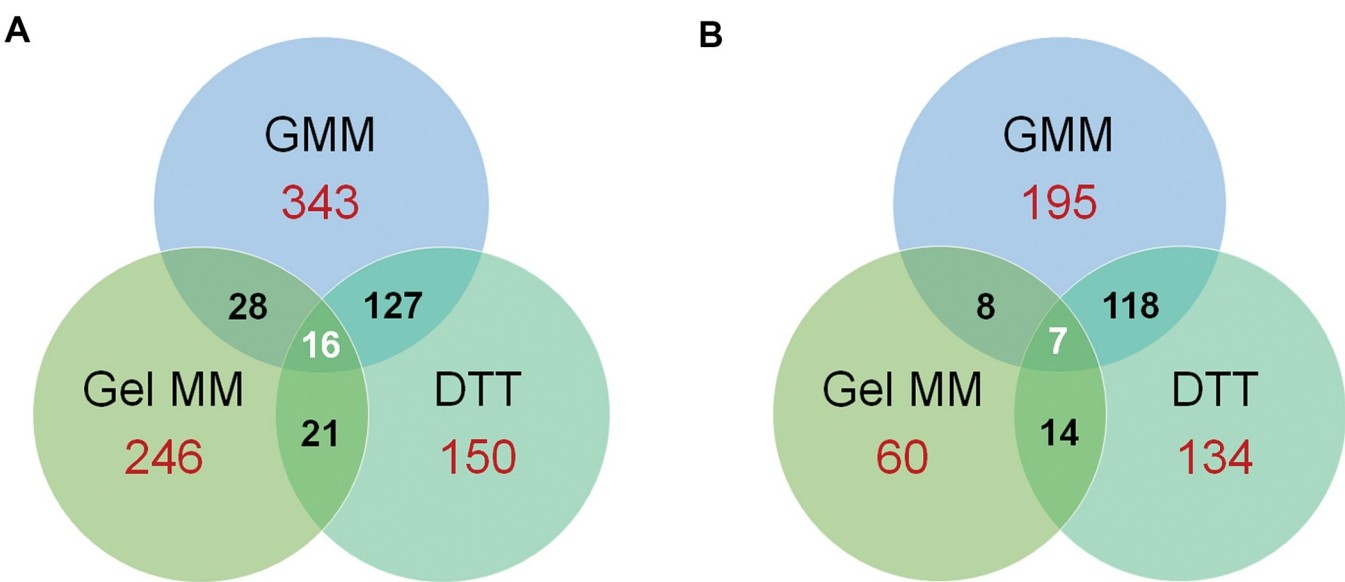

**Fig 9. Unique HacA-dependent transcriptomes across three environments.** Venn diagrams summarizing the number of unique and overlapping *ΔhacA* DEGs under the three tested conditions (A) down-regulated in *ΔhacA* relative to WT, (B) up-regulated in *ΔhacA* relative to WT.

between the two environments. We, therefore, postulate that HacA output in this organism is non-binary (e.g. bUPR vs iUPR), but is rather tunable based on integrated pathway inputs, including from nutrient signaling pathways, that modulate HacA stability or target binding post-translationally. Such regulation has been observed in the mammalian system, where acetylation/de-acetylation influences the protein stability and transcriptional activity of the HacA ortholog, Xbp1 [44]. Regarding *A. fumigatus*, Krishnan and colleagues revealed a form of post-transcriptional regulation whereby ER stress leads to transcript-specific translational induction via polysome docking [19]. Interestingly, the *hacA* transcript itself is not apparently regulated through this mechanism, and thus it remains unclear how hacA/HacA activity levels might be regulated beyond the level of mRNA splicing. It is possible, on the other hand, that many of the DEGs observed in *ΔhacA*, relative to WT, are not directly regulated by the HacA transcription factor itself or one of its transcription factor targets. For example, Weichert et al. demonstrated that 1) HacA regulates calcium ion homeostasis in the ER through the regulation of the Ca2+ ATPAses—*srcA* and *pmrA*, 2) the loss of such regulation results in $Ca^{2+}$ dyshomeostasis and a hypersensitivity to calcineurin inhibitors [16]. These data suggest that calcineurin signaling may be indirectly dependent upon UPR disruption, and its influence on its own downstream transcription factors (e.g. CrzA), could be influencing the transcriptome in the *ΔhacA* background. Ongoing studies in our group seek to understand if nutritional changes in the media quantitatively or qualitatively influence HacA binding targets and, if so, what sort of post-translational modifications or target-gene epigenetics are at play.

It is important to note that while our experiments seek to expand upon the microarray study of Feng et al. [15], in which the basal and induced UPR of *A. fumigatus* was originally described, there are key differences in the study design that may complicate a direct comparison. For example, Feng et al. defined ER-stress responsive genes as those that were shared among DTT and tunicamycin treatment groups (n = 57); bUPR and iUPR genes were furthermore defined as those altered in both the *ΔhacA* and *ΔireA* mutants [15]. Our study, by comparison, is less stringent in that it employs only a single canonical ER stressor (DTT) and one UPR mutant (*ΔhacA*). Coupled with our use of a more sensitive mRNA detection technique

(RNA-seq), we recognize that our view *A. fumigatus* UPR is markedly expanded. Another important consideration is the culturing conditions utilized in the experiments. Feng et al. used a rich medium (containing yeast extract and glucose) as the reference condition in which the bUPR was defined [15]. By contrast, and because our study included nutrition as a key variable, we employed a minimal medium where the carbon and nitrogen source varied between GMM (glucose/nitrogen) and Gel-MM (gelatin) while the micronutrient composition and pH remained constant between conditions. Based on the reduced mycelial outgrowth and enhanced conidiation of the WT on GMM relative to rich media (observable in Fig 1), we predict the former is inherently more stressful and may impart higher levels of baseline UPR activity. This may account for why *bipA* and other chaperones emerged as part of the bUPR in our study, but not in the Feng et al. report [15]. Along the same lines, the difference in media may influence the magnitude or even quality of the response to DTT. Finally, and perhaps most importantly, is the issue of intraspecies strain heterogeneity. The observations that 1) *ireA* is essential in Af293 (this study) but not in AfS28 (Feng et al.), and 2) the virulence defect of Af293 *ΔhacA* is more severe in a comparable model of IPA, suggest an expanded role of the UPR in the Af293 background [15]. Similar observations concerning orthologous gene function have been described previously. For example, loss of the calcineurin-regulated transcription factor, CrzA, similarly results in markedly stronger baseline growth defects in the Af293 compared to CEA10 [45,46] and, more strikingly, the VeA transcription factor plays opposing roles as a negative and positive regulator or condition in Af293 and CEA10, respectively [47,48]. Along those lines, Colabardini et al recently demonstrated a qualitatively distinct transcriptional signature between Af293 and the CEA10 derivative (CEA17) to caspofungin, which suggests the response to the conditions we have tested may also be variable across isolates [35,49]. Despite the myriad of technical and biological variables that distinguish the *A. fumigatus* UPR transcriptome studies, they both nevertheless support a conserved role for the pathway in regulating *A. fumigatus* oxidoreductive metabolism, mitochondrial function, hydrolase activity, and cell wall homeostasis beyond canonical protein folding and secretory genes.

Our observation that *ΔhacA*-infected corneas are sterile at 72 h p.i. and do not otherwise show signs of disease development, such as corneal opacification caused by infiltrating immune cells, suggests that the UPR is essential for the initiation of invasive growth in the corneal stroma. We reasoned that this is attributable, in part, to a critical loss of extracellular protease activity needed to obtain macronutrients from or penetrate through the collagen-rich stromal ECM. Our finding that the *ΔprtTΔxprG1ΔxprG2* mutant is fully virulent in the cornea, however, does not support this interpretation. It has been demonstrated that the *ΔprtTΔxprG1* mutant is fully virulent in the lung as well, which puts into question the general importance of *A. fumigatus* proteases in promoting invasive growth, despite their demonstrable upregulation within infected tissue. Our analysis of the WT transcriptome reveals a broad upregulation of proteases in gelatin along with other hydrolytic enzymes (e.g. lipases), which likely reflects a broad de-repression of alternative carbon acquisition pathways. The *ΔprtTΔxprG1* mutant, therefore, is likely capable of acquiring non-proteinaceous nutrients from the ocular surface (e.g. lipids or carbohydrates in the tear film), or in the lung, in order to initiate growth [50–52]. Regarding the cornea, we predict that the release of matrix metalloproteases by fibroblasts and infiltrating neutrophils can both nutritionally supplement the fungus at later stages of disease and drive tissue damage [53,54]. It thus remains unclear if macronutrient starvation is principally involved in the avirulence of *ΔhacA*. We predict that it is, in some sense, in that the broad upregulation of secreted hydrolases in the tissue, whether they truly support growth or not, drives the development of unchecked ER stress in the mutant and stalls growth before it is cleared from the ocular surface.

In agreement with previous reports, we have shown *ΔhacA* is hypersensitive to cell wall perturbating agents (e.g. Congo red), which likely reflects alterations in overall cell wall composition, and/or architecture. Indeed, Richie et al. reported reduced glucose content in both the soluble and insoluble cell wall fractions of *ΔhacA*, suggesting a reduction in both alpha- and beta-glucan [14]. While the generalized secretion defect of *ΔhacA* is thought to impact the delivery of cell wall-modulating enzymes to the hyphal apex, our transcriptomic data further suggest a role for HacA in influencing, directly or indirectly, the expression of cell wall-related genes. Among the underrepresented genes in *ΔhacA* include a myriad of endo- and exo-glucanase genes that may influence proper cell wall turnover and modification [55,56]. Thus, the dysregulation of these genes may account for the generalized growth defect of the mutant in GMM, the enhanced growth defect on gelatin, and an even more marked penetration defect through the dense stromal matrix *in vivo*. Notably, one of the underrepresented genes in the mutant was *ags3*, which plays a critical role in synthesizing alpha-glucan and masking the underlying and immunogenic beta-1,3-glucan [57,58]. Although we did not appreciate any signs of inflammation or disease development in *ΔhacA*-infected corneas, it is possible that the mutant was more efficiently recognized and cleared from the cornea by resident immune cell populations or early-infiltrating neutrophils at very early time points post-inoculation [59–61]. Alterations in the *ΔhacA* cell wall may furthermore impart a hypersensitivity to antimicrobial cell surface proteins that directly inhibit fungal growth (e.g. lysozyme, B-defensins) or influence the interaction of the fungus with host cells (e.g. complement) [62–66]. An assessment of how fungal UPR disruption influences cell wall antigenicity, host cell activation, and phagocytic uptake is being actively investigated by our group.

The UPR may also promote fungal resistance to other stresses encountered at the corneal surface or upon interaction with infiltrating leukocytes. For example, proteins that are both present in the tear film and are secreted by neutrophils, including lactoferrin and lipocalin, sequester iron and other essential metals that are required for fungal growth [67]. Indeed, the topical instillation of purified lactoferrin can inhibit fungal growth in the cornea, and ablation of *A. fumigatus* siderophore synthesis renders the fungus avirulent [68]. Our data demonstrate a down-regulation of several iron homeostasis genes in *ΔhacA* on glucose and gelatin media, including a siderophore transporter (AFUA_3G13670) and a ferric chelate reductase (AFUA_8G01310), and these changes correspond to hypersensitivity of Af293 and AfS28 *ΔhacA* to the iron chelator BPS [15]. Moreover, the altered expression of metal homeostasis seems to correspond with a much larger shift in the oxidoreductase category, which supports a role for HacA in regulating mitochondrial function, redox homeostasis and, consequently, optimal growth.

An intriguing inference from our RNA-seq data is that UPR influences a variety of secondary metabolic pathways, namely the fumagillin gene cluster, which is statistically downregulated in *ΔhacA* in both glucose and gelatin media. This mycotoxin irreversibly inhibits the methionine aminopeptidase (MetAP) type 2 enzyme that hydrolyzes the initial methionine (iMet) of newly synthesized proteins and consequently influences a myriad of signaling pathways and cellular processes in both microbes and mammals. As such, the compound has been considered for clinical use as an inhibitor of angiogenesis and tumor growth, as an antiviral and antiparasitic. Regarding its potential influence on *A. fumigatus* virulence, the fumagillin cluster is upregulated in a murine model of IPA and, *in vitro*, the compound is toxic to pulmonary epithelial cells and inhibits a variety of important neutrophil functions, including ROS production and degranulation [69–71]. While the role of fumagillin during FK remains unknown, we reason its downregulation in *ΔhacA* could influence rapid clearance from the corneal surface before invasive growth is initiated. Moreover, various genes in the gliotoxin gene cluster are variably altered in *ΔhacA* (although the cluster as a whole is only statistically

significant in DTT), and this metabolite has a demonstrable impact on virulence in a cortico-steroid model of IPA [72–75]. Although Leal et al. demonstrated that gliotoxin-deficient mutants are not impaired for growth in an immunocompetent model of FK [76], we reason that mycotoxin production may have a more marked influence on virulence in our steroid model in which the functionality of infiltrating leukocytes is already partially impaired. Follow-up work from our lab will include the direct measurement of fumagillin and gliotoxin production in the *ΔhacA* mutant as well as the virulence of mycotoxin deficient strains in our model.

This study adds fungal keratitis to a long list of infectious contexts in which the UPR regulates fungal virulence [15,77–79]. In the corn smut pathogen *Ustilago maydis*, for example, the *hacA* ortholog (*cib1*), is spliced upon the plant penetration and cib1 knockout is markedly attenuated for virulence on maize seedlings. Interestingly, the overexpression of the the *cib1* spliced form also attenuated virulence, emphasizing the importance of timing in UPR activation for the pathogenesis of *U. maydis* [80,81]. To the best of our knowledge, however, there has been no direct assessment of the UPR activation state of *A. fumigatus* or any other human fungal pathogen *in vivo*. Interestingly, our results suggest that the level of pathway activation in *A. fumigatus* is variable across host environments, with the $hacA^i/hacA^u$ splice ratios from corneal tissue resembling the GMM reference, and those from the lung resembling the DTT treatment reference. At face value, this suggests that the HacA-dependent transcriptome in the infected lung resembles the canonical iUPR; however, taking our transcriptomic data into consideration, we predict that there is fine-tuning of UPR readout even across environments that trigger *hacA* splicing. Why comparatively less *hacA* splicing is observed in the FK cornea remains unclear since many lung-relevant ER stressors also present in the cornea, including nutrient limitation and oxidative stress imparted by infiltrating neutrophils. Indeed, deletion of the *A. fumigatus* genes encoding superoxide dismutase or the ROS-sensitive transcription factor gene, *yap1*, results in a hypersensitivity to exogenous oxidizers and reduced fungal burden in a murine model of FK [76]. Moreover, and despite its proximity to ambient air, neutrophils furthermore drive the development corneal hypoxia during herpes simplex virus-1 keratitis, suggesting low oxygen may also be relevant during FK [82]. Even though the salient stress differences between the cornea and lung remain obscure, HacA nevertheless appears to essential for invasive growth in both tissues due to its role in nutrient acquisition, intracellular redox balance, metal ion homeostasis, and cell wall homeostasis. Importantly, this suggests the pathway could serve as a target for novel antifungals that can improve both visual and survival outcomes in patients afflicted by this important pathogen.

## Materials and methods

### Ethics statement

All animal studies were performed in accordance with the Association for Research in Vision and Ophthalmology (ARVO) guidelines for the use of animals in vision research and were approved by the University of Oklahoma Health Sciences Center Institutional Animal Care and Use Committee (Protocol: 20-060-CI).

**Fungal strains and growth.** *Aspergillus fumigatus* Af293 served as the WT background and all fungal strains generated for this study are listed in the **S1 Table**. The strains were maintained on a glucose minimal medium (GMM AT) containing 1% glucose, Clutterbuck salts, and Hutner's trace elements, 10 mM ammonium tartrate, pH = 6.5 [83]. Phenotypes of the strains were assessed on: yeast, peptone, and dextrose (YPD), containing 2% dextrose, 2% peptone, 1% yeast extract; gelatin minimal medium (Gel-MM), containing 1% gelatin, Clutterbuck salts, and Hutner's trace elements, pH = 6.5; 1% skim milk containing Clutterbuck salts, and

Hutner's trace elements, pH = 6.5 or; GMM supplemented with 1% BSA (GMM BSA) as the sole nitrogen source. For solid media assays, agar was used as a solidifying agent. For stress assays, $2x10^3$ of the indicated strains were spot inoculated onto media containing the indicated concentrations of brefeldin A (BFA, Sigma B7651), tunicamycin (Enzo #BML-CC104-0010), hydrogen peroxide, menadione, Congo Red (Sigma #C6767), Bathophenanthrolinedisulfonic acid disodium salt hydrate (BPS, Sigma #11890) and incubated at the temperature and duration as described. For liquid culture assays, optical density was measured at 530 nm. *Ex vivo* growth involved inoculation of $2x10^3$ conidia onto brain or lung tissue explanted from C57BL/6 animals. Statistical analysis of radial growth, biomass, or optical density across replicate samples was performed in GraphPad Prism 9.3.1. as described in the corresponding figure legends.

**Generation of *A. fumigatus* mutant strains.** For the generation of protoplasts, conidia were inoculated in GMM AT supplemented with 0.5% yeast extract for 10 h at 30˚C and the resulting biomass was digested with an enzyme cocktail containing: 5 mg/ml of Lysing Enzymes from *Trichoderma harzanium*, 5 mg/ml of Driselase, and 100 μg of chitinase in an osmotic medium (1.2 M MgSO4, 10 mM Sodium Phosphate Buffer pH 5.8) at 30˚C for 4h. DNA and CRISPR ribonucleoproteins (RNPs) were introduced into protoplasts by co-incubation with polyethylene glycol (60% PEG M.W. 3350 w/v, Sigma #P4338) as previously described [84,85] and protoplasts were recovered on solid GMM AT media supplemented with 1.2 M sorbitol containing hygromycin (200 μg/mL), phleomycin (125 μg/mL) or pyrithiamine (1 μg/mL) as indicated.

Targeted deletion of the *hacA*, *xprG2*, and *xprG1* genes was performed using Cas-mediated homologous recombination as described previously [36]. Briefly, repair templates were generated by amplifying the *hph* (from pan7-1) or *bleR* markers (from p402R) using primers containing 35 bp homology to either side of the targeted locus. 10 μg of repair template DNA template were co-transformed into fungal protoplasts along with two Cas9 RNPs, each assembled with a CRISPR guide RNA (Integrated DNA Technologies) that targeted a PAM site at one end of the gene coding sequence. The *prtT* single deletion mutant was generated through a split-marker homologous recombination approach [86] in which ~1 Kb of the sequence upstream of the *prtT* coding sequence was fused to the first two-thirds of the *bleR* cassette (left arm construct) and the second two-thirds of the *bleR* cassette was fused to ~1 Kb of the downstream *prtT* flanking sequence (right arm construct). 10 μg of both the left and right arm constructs were then introduced to WT Af293 protoplasts.

To generate the *prtT/xprG1/xprG2* triple mutant, cross-reactive *xprG1/2* RNPs, along with a pyrithiamine (*ptrA*) repair template (also containing homology to both loci), was introduced into the *ΔprtT* protoplasts. PCR of genomic DNA was used to confirm the loss of the targeted locus among drug resistant transformants.

Complementation of *ΔhacA* was achieved through co-transformation of *bleR* and *hacA* (promoter + coding sequence) PCR products into mutant protoplasts. Ectopic integration of the *hacA* fragment among phleomycin-resistant transformants was confirmed by PCR of genomic DNA.

Disruption of the native *hacA* and *ireA* promoters with the tetracycline repressible (tTA/TetOff) cassette was performed in an Af293 mutant constitutively expressing *mCherry* and as previously described [36,38]. Briefly, the TetOff cassette amplified from pSK-606 was fused to a *bleR* amplicon by overlap PCR with primers that introduced 35 bp flanking homology to the target locus. The replacement construct was introduced into fungal protoplasts along with an RNPs that targeted the native locus. Homologous integration of the cassette was confirmed by PCR and phenotypes of the mutants were evaluated on GMM AT plates containing 80 μg/mL of doxycycline hyclate (Sigma #D9891).

All CRISPR guide RNAs and primers (Integrated DNA Technologies) used to generate transformation constructs and for genotyping are listed in the **S2 Table**.

**Azocollagen assay.** Collagenase secretion of *A. fumigatus* culture supernatants was quantified using Azocoll (Millipore Sigma) hydrolysis as previously described [87]. Briefly, conidia were inoculated at a concentration of 1 x $10^5$/mL in 25 ml GMM supplemented with 10% heat-inactivated fetal bovine serum (ATCC, USA) as shown previously [14]. Following culture at 200 rpm for 72 h at 35˚C, supernatants were centrifuged at 12,000 rpm for 5 minutes and a 100 μL aliquot was added to 400 μL of pre-washed Azocoll (5 mg/mL) prepared using a buffer containing 50 mM Tris-HCl (pH 7.5), 0.01% (wt/vol) sodium azide and 1 mM $CaCl_2$ [87]. The samples were then incubated at 37˚C for 3 h with gentle shaking. The tubes were centrifuged at 10,000 rpm for 3 min, and the absorbance was measured at 520 nm to measure the release of azo dye from the supernatant. These values were normalized to the dry weight (g) of the different strains using the 72 h biomass that was transferred to Whatmann paper and incubated at 56˚C for 48 h.

## Murine model of keratitis

*Inoculum*: 5x$10^6$ conidia were inoculated into 20 ml YPD and incubated for approximately 4 h at 35˚C until they reached the stage where most of the conidia were swollen. The biomass was then collected by centrifugation, washed 4x with PBS, and resuspended in 500 μL of PBS. The volumes were then adjusted to normalize the strains to an OD 360 nm of 0.8.

*Infections*: 6-8-week-old C57B6/6J mice (Jackson Laboratory) or CD-1 mice (Charles River) were used as indicated. Animals were immunosuppressed with an intraperitoneal (IP) injection of 100 mg/kg Depo-Medrol (Zoetis, USA) on the day preceding inoculation (day -1). On day 0, mice were anesthetized with 100 mg/kg ketamine and 6.6 mg/kg xylazine IP and an Algerbrush II was used to abrade the central epithelium of the right eye to a diameter of 1 mm. 5 μL of fungal inocula (described above) were applied to ulcerated eyes and removed with a Kim wipe after 20 minutes. A single dose of Buprenorphine SR (1 mg/kg) was administered subcutaneously for analgesia. The contralateral eye of each animal remained uninfected in accordance with the ARVO ethical guidelines.

*In vivo imaging*: Micron IV slit-lamp imaging: The animals were monitored every day p.i. by capturing live corneal photographs obtained from mice anesthetized with isoflurane with a Micron IV slit-lamp imaging system (Phoenix Research Labs Inc., CA, USA) for up to 72 h p.i. Anterior segment spectral-domain optical coherent tomography (OCT): The corneal thickness was measured for all the mice at 48 and 72 h p.i. using an OCT equipment (Leica Microsystems, IL, USA). Before imaging, the mice were anesthetized using isoflurane and a 12 mm telecentric lens was used to generate a 4x4 mm image. The reference arm was calibrated and set to 885 by the manufacturer. The InVivoVue driver software was used to analyze the images. To quantify the corneal thickness, measurement was performed using an 11x11 spider plot to cover the entire eye, and an average of 13 readings were taken, and analyzed by Ordinary one-way ANOVA in GraphPad Prism 9.3.1. Algerbrushed sham-infected eyes were used as a control.

*Histopathological examination*: Control and infected eyes were harvested and fixed with 10% neutral buffered formalin for 4 h followed by 70% ethanol until further processing. The eyes were sectioned (5 μm thick) and stained with Periodic acid Schiff-hematoxylin (PASH) to evaluate the fungal cell wall and the host inflammatory response in the murine model of FK.

*Fungal burden determination*: Corneas were dissected aseptically and homogenized in 1 ml of 2 mg/ml collagenase buffer, and a 100 μl aliquot was plated in triplicate onto inhibitory mold agar (IMA) plates and incubated at 35˚C for 24 h to determine the number of colony-

forming units (CFU) per cornea. The statistical analysis on the colony counts was performed across the group using Ordinary one-way ANOVA in GraphPad Prism 9.3.1. The sham-infected eyes were used as controls.

*Clinical scoring*: The micron images were visually scored using the criteria previously established for keratitis by two blinded reviewers [88]. Briefly, eyes were graded on a scale of 0 to 4 and averaged based on surface regularity, area of opacification, and density of opacification. The statistical analysis on the average score per cornea was measured across the group using Ordinary one-way ANOVA in GraphPad Prism 9.3.1.

*RNA isolation from infected corneas*: Animals were euthanized by $CO_2$ asphyxiation at the indicated time points p.i., and corneas were resected and stored at -80˚C until processing. Tissues were homogenized with a TissueLyser LT (Qiagen, Maryland, USA) at 50 oscillations/s for two 30 s cycles in the presence of stainless-steel beads (Green bead lysis kits, Next Advance, New York, USA). Total RNA was extracted from homogenate supernatants using an Rneasy kit (Qiagen, USA) per the manufacturer's guidelines.

## Murine model of invasive pulmonary aspergillosis

For the cumulative mortality study, 6–8-week-old CD-1 male mice were immunosuppressed using triamcinolone acetonide (40 mg/kg of body weight of Kenalog-10; Bristol-Myers Squibb) on days -1, +3, +5 and +7. Under anesthesia by isofluorane, $2 \times 10^6$ conidia from the indicated strains were instilled intranasally in a total volume of 40 μL PBS. Survival was monitored for 14 days.

For the histopathological experiment, 6–8-week-old CD-1 male mice were immunosuppressed using triamcinolone acetonide (40 mg/kg of body weight of Kenalog-10; Bristol-Myers Squibb) on days -1, +2, and +5. On days +1, +2, +3 and +5 p.i., animals were euthanized by $CO_2$ asphyxiation, and lungs were removed and fixed with 10% neutral buffered formalin. Sections of the lungs were stained with Grocott methenamine silver (GMS) or hematoxylin and eosin (H&E). Olympus virtual slide scanner was used to scan and image the samples.

*Isolation of total RNA from infected lungs*: Animals were euthanized by $CO_2$ asphyxiation at the indicated time points p.i. and lungs were resected and stored at -80˚C until processing. The frozen samples were then subjected to a lyophilization for 72 h and homogenized with TissueLyser LT (Qiagen, Maryland, USA) in the presence of 6.35 mm diameter stainless steel bead (BioSpec #11079635c). Total RNA was extracted from homogenate supernatants using an Rneasy kit (Qiagen, USA) per the manufacturer's guidelines.

## Transcriptomics

8 ml of GMM AT and gelatin MM were inoculated with WT or *ΔhacA* with a density of $1.0 \times 10^5$/mL in 6-well plates and incubated in 35˚C static culture for 48 h. As indicated, GMM AT cultures were treated with 10 mM DTT 2 h prior to harvest. The biomass was harvested and placed in a 1.5 mL microcentrifuge tube with 1.0 mm diameter glass beads (BioSpec, Bartlesville, OK, USA) and tissue was homogenized using TissueLyser LT (Qiagen, Maryland, USA). RNA extraction was performed using an Qiagen Rnease kit and treated with DNAse as described above. Stranded RNA-seq libraries were constructed using NEBNext poly(A) mRNA isolation kit along with the SWIFT RNA Library Kit (NEB, MA, USA) according to the established protocols at Laboratory for Molecular Biology and Cytometry Research (LMBCR) at The University of Oklahoma Health Sciences Center (OK, USA). The library construction was done using 100 ng of RNA. Each of the 18 libraries was indexed during library construction to multiplex for sequencing. Libraries were quantified using a Qubit 4 fluorometer (Invitrogen, MA, USA) and checked for size and quality on Bioanalyzer 2100 (Agilent, CA, USA).

Samples were normalized and pooled onto a 150 paired end run on NovaSeq (Illumina, CA, USA) to obtain 50M reads per sample. Differentially expressed genes (DEGs) between groups were considered to be significant if the difference was at least ≥4-fold. Gene ontology (GO) and Functional Catalogue (FunCat) were used to systemically categorize the DEGs.

## RT-PCR and qRT-PCR analyses

RNA isolated from fungal culture or tissue was Dnase-treated using the Dnase I kit (Millipore Sigma, Massachusetts, USA). The quantity and quality of RNA were determined using the Nanodrop 2000 (Thermo Fisher Scientific, Massachusetts, USA). The RNA was normalized for cDNA conversion using ProtoScript II First Strand cDNA Synthesis Kit (New England Bio-labs, Massachusetts, USA) following the manufacturer's protocol. qRT-PCR was performed using Luna Universal qPCR master mix (SYBR green; NEB, USA) on the QuantStudio 3 Real-Time PCR System (Thermo Fisher Scientific, Massachusetts, USA) and analysis was performed using QuantStudio Design and Analysis Software v1.5.2. The fold-expression changes were calculated using the 2-ΔΔCt method and followed by analysis using MS-Excel and Graph Pad Prism 9.3.1.

*hacA splicing*: To visualize the presence of the two *hacA* splice forms (uninduced form hacA$^u$– 665 bp; induced form hacA$^i$– 645 bp) from cDNA derived from fungal culture or infected tissues (as described above), end point PCR was performed using primers that spanned the predicted splice region (689/690 in **S2 Table**). The resulting amplicons were then run on a 3% agarose gel at 40 mV for 10 h and visualized with ethidium bromide staining. For quantitative analysis, qRT-PCR was performed with separate PCR reactions with primer sets that were specific to either the unspliced or spliced *hacA*.

All endpoint and qPCR primers used in this study are listed in the **S2 Table**.

## Supporting information

**S1 Fig. Genotypic validation and additional phenotypes of the Af293 *ΔhacA* mutant.** (A) An overview of the *hacA* deletion strategy using *in vitro* assembled Cas9 RNPs confirmed by PCR of genomic DNA, (B) 72 h colony diameters of the indicated strains on YPD, GMM with ammonium tartrate as the nitrogen source (GMM-AT), GMM with BSA as the nitrogen source (GMM-BSA), or gelatin minimal medium. Data represented as mean of triplicate samples analyzed by two-way ANOVA **** <0.0001, (C) Colonial appearance of the indicated strains following growth on varying concentrations of brefeldin A (BFA) for 35˚C for 48 h. (TIF)

**S2 Fig. IreA is essential for the growth of *A. fumigatus* (Af293).** (A) Schematic representation of *hacA* and *ireA* promoter replacement with the Tet-Off cassette tagged with the phleo-mycin resistance gene (*bleR*), (B) PCR strategy and genotyping of the generated $Ptet_{off}$-*hacA* and $Ptet_{off}$-*ireA* strains, (C) Colonial appearance of the indicated strains grown with or with-outh 80 µg/mL doxycycline. Photographs taken following transfer of hyphal plugs onto the media and incubating at 35˚C for 72 h. (TIF)

**S3 Fig. *A. fumigatus ΔhacA* is avirulent in the cornea of both male and female C57BL/6J mice.** (A) Representative slit-lamp images of infected murine corneas at 24, 48 and 72 h p.i. (B) 72 h clinical disease scores of pooled male and female animals; data analyzed by Ordinary one-way ANOVA **** <0.0001. (C) 72 h corneal thickness measurements based on OCT and measuring 13 points across cornea; (n = 10/group, analyzed by Ordinary one-way ANOVA p-value **** <0.0001. (D) Fungal burden at 72 h p.i. as determined by colony forming unit

(CFU) analysis from corneal homogenates; data analyzed by Ordinary one-way ANOVA p-value **** <0.0001; ** 0.0012. (E) Representative PASH stained corneal corneal sections at 72 h p.i. (F) The same 72 h clinical disease scores in panel B with male and female animals separated. (G) The same 72 h corneal thickness measurements in panel C, with male and female animals separated.
(TIF)

**S4 Fig. Genotypic validation and collagenase activity of the protease-deficient mutants.** (A) Genomic PCR demonstrating replacement of the *prtT* coding sequence with the *bleR* cassette. (B) Genotypic analysis of the *ΔxprG1ΔxprG2*, *ΔxprG2*, and *ΔprtTΔxprG1ΔxprG2* mutants using a multiple PCR scheme. (C) Collagenase activity of cultured supernatants following 72 h incubation at 35˚C in GMM-FBS; data are depicted as raw absorbances at 520 nm following azocoll degradation and represent the mean of triplicate samples analyzed by Ordinary one-way ANOVA * p-value <0.0001 compared to the negative control. (D) Amino Acid sequence alignment for *xprG1* (Afu8g04050) and *xprG2* (Afu1g00580) using a Smith-Waterman local alignment on SnapGene v7.0.2. The alignment shows NDT80 DNA binding domain (amino acid residues 86–290) in XprG1 (green) and the putatively non-functional NDT80 (missing amino acid residues 235–270) domain of XprG2 sequence (red).
(TIF)

**S5 Fig. Analysis of *ΔhacA* sensitivity to various stresses.** (A) Dry weight of Af293 WT and *ΔhacA* grown in liquid GMM with and without 2 μM $H_2O_2$ for 72 h at 35˚C; data represent the mean of triplicate samples analyzed by unpaired T-test. (B) Conidia of WT and *ΔhacA* were spotted on GMM containing a concentration gradient of $H_2O_2$. Photographs taken after 72 h incubation at 35˚C. (C) Dry weight of Af293 WT and *ΔhacA* grown in liquid GMM with and without 60 μM menadione for 72 h at 35˚C; data represent the mean of triplicate samples analyzed by unpaired T-test. (D) Photographs taken of the indicated strains following growth on GMM and gelatin MM at 35˚C for 72 h in normoxia or hypoxia (1% $O_2$). The *ΔhacA* mutant is developmentally delayed on gelatin, but its mycelial growth is indistinguishable in normoxia and hypoxia. (E) 24–72 h growth rate measurement on YPD plates with 200 μM of the iron chelator bathophenanthrolinedisulfonate (BPS). (F) Conidia of the indicated strains were spotted onto GMM a concentration gradient of BPS; photographs taken after 72 h incubation at 35˚C. (G) Conidia of the indicated strains were spotted onto GMM a concentration gradient of Congo red; photographs taken after 72 h incubation at 35˚C.
(TIF)

**S1 Data. DEGs and enriched terms for WT-GMM vs WT-DTT.**
(XLSX)

**S2 Data. DEGs and enriched terms for WT-GMM vs WT-Gel MM.**
(XLSX)

**S3 Data. DEGS and enriched terms for WT vs *ΔhacA* (GMM).**
(XLSX)

**S4 Data. DEGS and enriched terms for WT vs *ΔhacA* (Gel MM).**
(XLSX)

**S5 Data. DEGS and enriched terms for WT vs *ΔhacA* (DTT).**
(XLSX)

**S6 Data. Common *ΔhacA* DEGs across all three conditions (GMM, Gel MM, DTT).**
(XLSX)

**S7 Data. Unique *ΔhacA* DEGs for each of the three conditions (GMM, Gel MM, DTT).**
(XLSX)

**S1 Table. Strains and plasmids used in the study.**
(XLSX)

**S2 Table. Primers and CRISPR RNAs used in this study.**
(XLSX)

## Acknowledgments

We thank Mark Dittmar and staff (DMEI Animal Research Facility), Linda Boone and Louisa Williams (DMEI Imaging Core), and the Institutional Research Core Facility at OUHSC which provided total RNA library construction and sequencing. We also thank Dr. Jarrod Fortwendel at the University of Tennessee Health Sciences Center for the generous gift of the pSK-606 plasmid.

## Author Contributions

**Conceptualization:** Manali M. Kamath, Jorge D. Lightfoot, Kevin K. Fuller.

**Data curation:** Manali M. Kamath, Kevin K. Fuller.

**Formal analysis:** Manali M. Kamath, Jorge D. Lightfoot, Ryan M. Kiser, Kevin K. Fuller.

**Funding acquisition:** Kevin K. Fuller.

**Investigation:** Manali M. Kamath, Jorge D. Lightfoot, Emily M. Adams, Ryan M. Kiser, Becca L. Wells, Kevin K. Fuller.

**Methodology:** Manali M. Kamath, Jorge D. Lightfoot, Kevin K. Fuller.

**Project administration:** Kevin K. Fuller.

**Resources:** Kevin K. Fuller.

**Supervision:** Kevin K. Fuller.

**Validation:** Manali M. Kamath, Kevin K. Fuller.

**Visualization:** Manali M. Kamath, Emily M. Adams.

**Writing – original draft:** Manali M. Kamath, Kevin K. Fuller.

**Writing – review & editing:** Manali M. Kamath, Jorge D. Lightfoot, Emily M. Adams, Ryan M. Kiser, Kevin K. Fuller.

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
