## [Decision Letter · Decision Letter 0]

19 Jun 2023

Dear Assistant Professor Fuller,

Thank you very much for submitting your manuscript "Basal UPR activity in Aspergillus fumigatus regulates adaptation to nutrient stress and is critical for the establishment of corneal infection" for consideration at PLOS Pathogens. As with all papers reviewed by the journal, your manuscript was reviewed by members of the editorial board and by several independent reviewers. In light of the reviews (below this email), we would like to invite the resubmission of a significantly-revised version that takes into account the reviewers' comments.

Specifically, all three reviewers raised concerns with the presentation and interpretation of the presented data. A significant rewrite here taking into consideration their excellent suggestions will help clarify the conclusions, and their rigor, being drawn from the data. With respect to the latter, Reviewer 1 makes an excellent point about the tissue specificity of the virulence defect. This finding is perhaps the most significant and novel of the study and warrants the additional experiments suggested by this reviewer (i.e. an IPA murine model). Results here may also help with presentation of the data and interpretation of the nutritional phenotypes being observed. 

We cannot make any decision about publication until we have seen the revised manuscript and your response to the reviewers' comments. Your revised manuscript is also likely to be sent to reviewers for further evaluation.

Sincerely,

Robert A. Cramer, PhD

Academic Editor

PLOS Pathogens

Alex Andrianopoulos

Section Editor

PLOS Pathogens

Kasturi Haldar

Editor-in-Chief

PLOS Pathogens

orcid.org/0000-0001-5065-158X

Michael Malim

Editor-in-Chief

PLOS Pathogens

orcid.org/0000-0002-7699-2064

Reviewer's Responses to Questions

**Part I - Summary**

Reviewer #1: This study examines the impact of the UPR on the ability of A. fumigatus to cause fungal keratitis. The results demonstrate that a mutant lacking the UPR transcription factor HacA is unable to establish a corneal infection. This predicts that the fungus is under ER stress in the corneal environment, which necessitates UPR activation to restore homeostasis. Surprisingly, the authors found that hacA splicing, and the expression of UPR target genes, was not induced during corneal infection relative to levels seen during optimal growth conditions in vitro. However, protease-encoding genes were upregulated in vivo, consistent with the ability of the fungus to use the high collagen content of the cornea as a substrate. The hacA mutant was deficient in protease secretion, which correlated with poor growth on protein-rich substrates. Using gelatin medium as a model of the collagen-rich cornea, the study revealed a role for hacA in the expression of genes involved in metabolic processes that may facilitate nutrient versatility. These findings suggest that the corneal environment does not trigger acute ER stress in A. fumigatus, but that basal activity of the canonical UPR is nonetheless required for infection.

Major strengths

• The manuscript is well written and an interesting read.

• Fungal keratitis has a high rate of morbidity and is an understudied area in fungal biology. The findings raise the possibility that topical applications of emerging UPR inhibitors could have therapeutic potential for keratitis caused by this, as well as other filamentous fungi such as Fusarium.

• A major strength of the paper is the demonstration that the hacA mutant is incapable of causing fungal keratitis and is cleared within 72 h. This is an interesting contrast to previous studies showing that a hacA mutant has attenuated disease progression in a model of invasive pulmonary aspergillosis, but still capable of causing disease. This indicates a tissue-specific requirement for the UPR during infection.

Reviewer #2: The manuscript by Kamath et al. addresses the role(s) of UPR activity in regulating nutrient stress as it pertains to Aspergillus fumigatus infection in the murine corneal infection model. The study is well-designed, and the manuscript is well-written. Authors have clearly established that, like UPR activation is indispensable in a murine model of invasive pulmonary aspergillosis, hacA is essential for the establishment of infection in a model of fungal keratitis . However, major inferences of the study are not supported by data or are over-extrapolated.

Reviewer #3: The authors have presented an interesting piece of work in an understudied yet important infectious disease. Fungal keratitis affects almost 1.5 million eyes each year resulting in over 500k blind eyes. This is one of very few research outputs that has sought to uncover specific mechanisms of virulence associated with this type of infection which is significantly different from other diseases caused by A. fumigatus.

The authors present convincing evidence that the transcription factor HacA, which regulates the unfolded protein response (UPR) in A. fumigatus, plays a critical role in corneal infections. Notably and surprisingly they show that the hacA transcript, which is processed from an uninduced form (hacAu) to an induced form (hacAi) under conditions that induce UPR (high temperature and exposure to the reducing agent DTT) is not induced beyond basal levels (GMM) in a corneal infection model, or in the presence of gelatin -a proxy for corneal collagen which is presumed to be the key source of nutrition for Af in the eye. The authors surmise that the lack of virulence in the hacA null is related to a regulatory role carried out by basal levels of the induced form of hacA. In support of this, transcriptional data is presented that shows loss of HacA results in downregulation of a number of genes associated with the UPR in standard culture conditions and in gelatin MM (although this is not directly stated). Overall the indication from the data is that migration of functional, correctly folded proteins through the ER in environments where secreted proteins are needed for nutritional acquisition require functional HacA. Although it is clear that hacA is required for growth in proteinaceous growth media, it is not responsible for regulation of collagenases and proteases that are upregulated upon exposure to the cornea, providing further circumstantial evidence that export but not production of these enzymes is affected. The authors are unable to demonstrate that virulence can be rescued by supplementation of AF inoculated eyes with glucose leaving them to speculate as to the cause of the lack of virulence in the hacA null mutant.

**Part II – Major Issues: Key Experiments Required for Acceptance**

Reviewer #1: 1. The sterilization of the mutant within 72 contrasts the attenuated virulence of this mutant in a model of invasive pulmonary aspergillosis, suggesting an important tissue-specific virulence defect. However, an alternate explanation is that the difference is due to variations in the genetic background of the fungus and/or the use of more recent technology for gene deletion (the IPA study was published in 2009). An experiment showing that the newer mutant developed in this study can still cause disease in a pulmonary model would clearly demonstrate that UPR support of corneal infection is context-specific and would strengthen the manuscript.

2. Another explanation for the inability of the mutant to infect the cornea is that it is hypersensitive to antimicrobial compounds in tear film (for example, the hacA mutant in Alternaria is hypersensitive to plant antimicrobial compounds). Does the hacA mutant show hypersensitivity to any of the major antimicrobial compounds found in tear film?

3. Line 353. “to our knowledge, there has been no direct assessment of whether any fungal pathogen activates the UPR in host tissue”. This should be corrected since activation of the UPR in vivo has been shown at least for cib1 (hacA homolog) in Ustilago Maydis (Plant cell 2010, mBio 2019). Since hacA is not induced in the cornea, this would support a model in which the impact of the UPR on virulence is tissue-specific, which is interesting and has important implication for future therapies that target the pathway.

Reviewer #2: . Some of the major concerns are listed below:

1. The primary inferences in the manuscript are based on the RNA data as investigated by transcriptomics or RT-PCR, or qRT-PCR. However, transcriptomic studies only provide a snapshot of RNA levels at a given time point. Thus, mRNA dynamics may not correlate with protein levels.

2. Authors argue that bUPR and iUPR may represent two ends of the UPR spectrum, and their corneal model is a source of sub-acute ER stress. This was not shown with changes in Hac-induced levels that were similar to basal levels. This was inferred based on changes in the transcriptome, which suffers from the argument in point 1.

3. Authors have mainly argued hacA-induced levels as the sub-acute level of ER stress. However, they fail to discuss the hacA independent role(s) of sensor ireA (PMID 22028661). How does basal and/or subacute ER stress (if it exists) regulate ireA and hacA independent functions?

4. Also, there is a direct link between the calcineurin pathway and ER stress response in baker’s yeast. How calcineurin and other known pathways of ER stress play a role are not mentioned/discussed.

5. Another major argument is the increased expression of major collagenase genes (Fig1b). How is that related to virulence? A �prtT mutant deficient in the secretion of these major proteases showed ex vivo defects but was indispensable for virulence. If the changes in the secretory pathway induce ER stress that is essential for infection and if these major proteases are involved in infection, authors should discuss the prtT data (PMID 19564385). Additionally, as glucose supplementation did not change corneal infection, the role of Er stress-dependent protease expression is not clearly defined.

6. As the authors discussed the multifactorial role(s) of hacA in nutrient stress, secondary metabolism, and other growth features (cell wall homeostasis), authors need to take into account that hacA mutant showed 50% mortality compared to the control. Thus, their aim of finding inhibitors that can be used to treat both IPA and FK still needs more work. As these survival/mortality analyses differ with the immune suppression, the major secondary metabolite gliotoxin might be playing a role; however, the authors fail to discuss the role(s) of immune suppression.

7. Authors should be careful in drafting the manuscript. Line 421, “Pearlman paper,” should be properly cited.

Reviewer #3: While the experimental work seems to be robust and does generally support the hypothesis I have a number of significant issues with the overall narrative.

Ultimately the information as presented is very confusing. Several things contribute to this:

1. Some titles are contrary to the data presented e.g. Transcriptomics supports distinct roles for the UPR under acute ER and nutritional stress. In this section the transcriptomic data does not seem to support any major role for the UPR in ‘nutritional stress’. Although I do accept there is a role for UPR during growth in nutritional stress (as there is for growth per se)– the requirement for hacA on proteinaceous media is not presented until much later in the manuscript.

2. Line 239 states that secretion of collagenases was impaired in the mutant – at that point in the manuscript the authors had not established that loss of hacA was not involved in transcriptional regulation of these enzymes – and hence a conclusion could be drawn that production (not just secretion) of collagenases was affected.

3. The link between the requirement for protease activity and investigation of the UPR is not well made following the first results section.

A few other critical issues with the ms:

1. Table 1: the legend lacks any meaningful description of the data presented. Also Exact p-values are presented: Given the dataset presented it is more appropriate to present corrected p-values (Benjamini-Hochberg) to account for false discovery. I suspect that this will remove protein folding as a hit from the DTT data however it is sufficient just to state the specific genes affected in this group to show impact on protein folding.

2. The supplementation of nutrients in the animal infection model with glucose does not account for the fact that Af is probably using the proteinaceous component of the cornea as a nitrogen source. This should be addressed.

Overall I feel the narrative and logical flow of the manuscript needs addressing. This will require reordering of the data presented and highlighting the critical points of the work more clearly.

**Part III – Minor Issues: Editorial and Data Presentation Modifications**

Reviewer #1: • For the nutritional experiments, was the medium solidified with agar or agarose (this should be mentioned in the methods)? Agar provides trace nutrients that can confound nutritional experiments. This is not a major point, since there is a clear difference with the mutant regardless, but the authors may see a bigger difference if agarose is used.

• Line 20 – ‘completely’ is redundant with ‘unable’

• Line 163 – consider adding language to explain the difference between in vivo and in vitro baseline. i.e. “…beyond baseline, defined as in vitro conditions where the medium supplies reduced sources of carbon and nitrogen that do not require breakdown by secreted fungal hydrolases”.

• Line 263. Fig. 1 should be Fig. 3

• Line 421. Missing reference (Pearlman paper)

• Since many outside the field are not familiar with this model, I think it would be worthwhile to clarify early in the manuscript that the model creates an epithelial breach to allow the fungus to colonize the collagen-rich environment of the corneal stroma. I also think it would be worth emphasizing that the cornea is a unique environment relative to the lung, particularly in the context of vascularization and immune-privilege (which could account for why the findings for this mutant differ between cornea and lung).

Reviewer #2: Please superscript the divalent metal cations. The degree sign seems to be a superscript "O" and not the degree symbol.

Reviewer #3: Some minor issues

1. Here seems to be some inappropriate self-citation on line 224 – Lightfoot and Fuller seems to refer to transformation of Fusarium solani.

2. The legend for figure 1 is not sufficiently detailed. Panel (B) – assume the transcripts have been normalised to B-tubulin and then compared to growth in standard culture conditions? Panel D include timepoints.

3. Figure 2 is not contributing anything to the paper – remove.

4. Figure 3 E- legend lacking detail.

5. The references are numbered at the end of the document but named in the ms- this made review difficult.

6. Figure title for 7 is not appropriate.

7. Line 421 – ‘Pearlman paper’ needs reference.

All figure legends should be checked carefully to ensure the data is correctly described.

PLOS authors have the option to publish the peer review history of their article (what does this mean?). If published, this will include your full peer review and any attached files.

Reviewer #1: No

Reviewer #2: No

Reviewer #3: No
---

## [Decision Letter · Decision Letter 1]

14 Oct 2023

Dear Assistant Professor Fuller,

We are pleased to inform you that your manuscript 'The Aspergillus fumigatus UPR is variably activated across nutrient and host environments and is critical for the establishment of corneal infection.' has been provisionally accepted for publication in PLOS Pathogens.

Best regards,

Robert A. Cramer

Academic Editor

PLOS Pathogens

Alex Andrianopoulos

Section Editor

PLOS Pathogens

Kasturi Haldar

Editor-in-Chief

PLOS Pathogens

orcid.org/0000-0001-5065-158X

Michael Malim

Editor-in-Chief

PLOS Pathogens

orcid.org/0000-0002-7699-2064

Reviewer Comments (if any, and for reference):

Reviewer's Responses to Questions

**Part I - Summary**

Reviewer #1: The authors have done a nice job of addressing the first critique, including the performance of additional experiments. These revisions have improved the overall manuscript.

**Part II – Major Issues: Key Experiments Required for Acceptance**

Reviewer #1: None required

**Part III – Minor Issues: Editorial and Data Presentation Modifications**

Reviewer #1: None identified

PLOS authors have the option to publish the peer review history of their article (what does this mean?). If published, this will include your full peer review and any attached files.

Reviewer #1: No

---

## [Editor Report · Acceptance letter]

24 Oct 2023

Dear Assistant Professor Fuller,

We are delighted to inform you that your manuscript, "The Aspergillus fumigatus UPR is variably activated across nutrient and host environments and is critical for the establishment of corneal infection.," has been formally accepted for publication in PLOS Pathogens.

Best regards,

Kasturi Haldar

Editor-in-Chief

PLOS Pathogens

orcid.org/0000-0001-5065-158X

Michael Malim

Editor-in-Chief

PLOS Pathogens

orcid.org/0000-0002-7699-2064